## Dynamics of a Geomagnetic Storm on 7-10 September 2015 as Observed by TWINS and Simulated by CIMI

Perez<sup>1</sup>, Joseph D., James Edmond<sup>1</sup>, Shannon Hill<sup>2</sup>, Hanyun Xu<sup>1</sup>, Natalia Buzulukova<sup>3</sup>, Mei-Ching Fok<sup>3</sup>, Jerry Goldstein<sup>4,5</sup>, David J. McComas<sup>6</sup> and Phil Valek<sup>4,5</sup>

<sup>1</sup>Auburn University, Auburn, AL 36849, USA
<sup>2</sup>Emory University, Atlanta, GA 30322, USA
<sup>3</sup>NASA Goddard Space Flight Center, Greenbelt, MD 20771, USA
<sup>4</sup>Southwest Research Institute, San Antonio, TX 78228, USA
<sup>5</sup>University of Texas at San Antonio, San Antonio, TX 78249, USA
<sup>6</sup>Department of Astrophysical Sciences, Princeton University, NJ 08540, USA

Correspondence to: J. D. Perez, perez@physics.auburn.edu

Abstract. For the first time, direct comparisons of the equatorial ion partial pressure and pitch angle anisotropy observed by TWINS and simulated by CIMI are presented. The TWINS ENA images are from a 4-day period, 7-10 September 2015. The simulations use both the empirical Weimer 2K and the self-consistent RCM electric potentials. There are two moderate storms in succession during this period. In most cases, we find that the general features of the ring current in the inner magnetosphere obtained from the observations and the simulations are similar. Nevertheless, we do also see consistent contrasts between the simulations and observations. The simulated partial pressure peaks are often inside the observed peaks and more toward dusk than the measured values. There are also cases in which the measured equatorial ion partial pressure shows multiple peaks that are not seen in the simulations. This occurs during a period of intense AE index. The CIMI simulations consistently show regions of parallel anisotropy spanning the night side between approximately 6 and 8 R<sub>E</sub> whereas the parallel anisotropy is seen in the observations only during the main phase of the first storm. The evidence from the unique global view provided by the TWINS observations strongly suggests that there are features in the ring current partial pressure distributions that can be best explained by enhanced electric shielding and/or spatially-localized, short-duration injections..

**Key Words**. Magnetospheric physics (Storms and substorms, Magnetosphere configurations and dynamics) – Space plasma physics (charged particle motion and acceleration)

## **1** Introduction

The Earth's inner magnetosphere contains a large-scale current system, the ring current, in which the current is carried by trapped ions that are injected from the magnetotail and generally drift westward. It is a major contributor to magnetic depressions measured in the Earth's equatorial region that are expressed in terms of the Dst or SYM/H indices which characterize the time-evolution of geomagnetic storms. The plasma sheet is a primary source of particles in the inner magnetosphere. Therefore understanding and predicting the dynamics of the injected particles is a key factor in understanding the formation and decay of the ring current. This challenge can be addressed by a comparison of model and simulation results with observations.

There have been many studies which compared model results to observations. Kistler and Lawson (2000) used 2 different magnetic field models, dipole and Tsy89 (Tsyganenko, 1989), along with two different electric potential models, Volland (Volland, 1973)-Stern (Stern, 1975) and Weimer96 (Weimer, 1996), to calculate ion paths in the inner magnetosphere. They compared the results with in-situ proton energy spectra measured by the Active Magnetospheric Particle Tracer Explorers (AMPTE) (Gloeckler et al, 1985) over a range of local times. They found that, in the inner magnetosphere, the electric field has a much stronger effect on the particle paths than the magnetic field and that the Weimer96 model gave a better match to the features of the observed energy spectra than the Volland-Stern model. But the energy at which the drift paths became closed, 40-50 keV, was not in agreement with the observations. It is to be noted that the effects of induction electric fields were not included in this analysis. Angelopoulos et al. (2002) added co-rotation electric fields to Volland-Stern, Weimer 96, Weimer 2000 along with modifications to improve fits to instantaneous electric field measurements by POLAR/HYDRA (Scudder et al., 1995) and Defense Meteorological Satellite Program satellites to compare with in-situ measurements of ion spectrograms from POLAR/HTDRA, EQUATOR-S (Kistler et al., 1999) and FAST (Carlson, et al., 2001). They found differences that seemed to require the inclusion of local inductive electric fields and/or particle injections. Ebihara et al., (2004) modeled discrete energy bands observed by POLAR using a dipole magnetic field and a realistic electric field to show that changes in the convection electric field produced better results.

De Michelis et al (1999) obtained images of pressure in the equatorial plane, both orthogonal and parallel, and anisotropy using 2-year averages of proton distributions measured by AMPTE/CCE-CHEM (Dassoulas et al., 1985; Gloeckler et al., 1985). They located 2 current systems, the inner portion of the cross-tail current and the ring current during times of AE > 100 nT, and both the full and partial ring current along with region 2 currents for 100 nT 

in this region have been seen from ENA images for very strong storms (Fok et al, 2003).

# **5.2** Comparison of Equatorial Ion Partial Pressure Peaks and Anisotropies at Specific Times

The following subsections will examine in detail a number of specific times during these two storms in order to address similarities and differences in the simulations with an empirical and a self-consistent electric field model and with observations. One apparent difference in what follows is the magnitude of the equatorial partial pressure for the three cases. The maximum on the color bars for Figures 3-9 were chosen to be different for each time in order to emphasize the spatial dependence of the pressure distribution. The maxima for the two CIMI simulations are very similar, i.e., the RCM vary from 20-38 nPa and the Weimer 2K from 15-30 nPa. But the maxima of the TWINS peaks vary from 1-4 nPa, which is significantly smaller.

The magnitude of the ion intensities derived from the ENA images has been addressed in several previous comparisons with in-situ measurements. Vallat et al. (2004) compared Cluster-CIS (Réme et al., 2001) and IMAGE-HENA observations and found that for relatively strong fluxes, the agreement was excellent for two cases, but for another the ion flux determined from the ENA images was somewhat higher than the in-situ observations and in another it was significantly lower. Grimes et al. (2013) compared THEMIS (Angleopoulos, 2008) spectral measurements with spectra obtained from TWINS ENA images and found that the in-situ fluxes were a factor of 3 times greater than those obtained from the ENA images. Perez et al. (2016) compared 30 keV ion fluxes obtained from TWINS ENA images with in-situ measurements by RBSPICE-A (Mauk et al., 2013) and found good agreement in both the average time dependent trend and in the magnitude. The in-situ measurements, of course, showed more structure given their much higher spatial and temporal resolution. Goldstein et al. (2017) analyzed data from THEMIS, Van Allen probes, and TWINS for a large storm to find that the ion fluxes obtained from the ENA images were generally lower than those from the in-situ measurements. They also found significant variations in the in-situ data. So while some part of the difference in the partial pressures obtained from TWINS measurements and CIMI simulations are due to the larger energy range included in the CIMI pressures, it is not the entire explanation. The issue of the absolute magnitude remains an important, unresolved issue, but the fluxes obtained from ENA images have been shown to reflect the global structure of the trapped ring current particles, and that is the emphasis in this study.

#### 5.2.1 2200 UT 07 September 2015

Figure 3 shows the equatorial partial pressure profiles and the pressure anisotropy from the CIMI/RCM simulation, the TWINS observations, and the CIMI/Weimer 2K simulation at 2200 UT 07 September 2015. This was late in the main phase of the first storm (See Figure 1.). The radial locations of the peaks differ by less than 1  $R_E$ . The MLT locations of the partial pressure peaks, however, differ by 3 hours in MLT. While the TWINS peak is near midnight, the CIMI peaks are well into the dusk/midnight sector with the CIMI/Weimer even closer to dusk. Results

for the Weimer96 when compared with the RCM for a very strong storm showed even greater shielding for the RCM when compared to the empirical Weimer model (Fok et al., 2003). Note, however, that for this weaker storm, the MLT spread in the peaks of the partial pressure distributions do overlap. It is also to be noted that the TWINS results show more radial structure.

The pressure anisotropy shown in Figure 3 is defined as

$$A = \frac{P_{\perp} - P_{\parallel}}{P_{\perp} + P_{\parallel}}$$

with

$$\begin{cases} P_{\perp} \\ P_{\parallel} \end{cases} = 2\pi \int_{-1}^{+1} d\cos\alpha \begin{cases} \sin^2\alpha \\ 2\cos^2\alpha \end{cases} \left( \int_0^\infty dE \sqrt{2mE} \ F(E, n, \cos\alpha) \right) \end{cases}$$

where  $\alpha$  is the ion pitch angle, *E* is the ion energy, *n* is the ion density, *m* is the ion mass and *F*(*E*,*n*,*cos*  $\alpha$ ) is the number flux per unit area, energy, time, steradian. This definition is derived from Braginskii (1965) and is consistent with previous formulations, e.g., Lui et al. (1987).

The pressure anisotropy at the pressure peaks is somewhat perpendicular in all 3 cases. We also note a region of parallel anisotropy at  $R > 6-7 R_E$  from pre-midnight to dawn in all 3.

#### 5.2.2 0400 UT 08 September 2015

Figure 4 shows results for 0400 UT 08 September 2015 in the same format. This was early in the rapid recovery phase of the first minimum in SYM/H. (See Figure 1.) The radial location of the partial pressure peaks again differ by less than 1 R<sub>E</sub>. This time, however, all the peaks are in the dusk/midnight sector. Again the CIMI/Weimer 2K is closer to dusk than the CIMI/RCM pressure profiles. The TWINS peak is between the two simulations. The CIMI/Weimer 2K pressure distribution is more symmetric than the others even though the ASY/H shown in Figure 1 is > 50 nT. The region of parallel pressure anisotropy in the CIMI results does not appear in the TWINS results which are more nearly isotropic in general compared to the CIMI simulations.

#### 5.2.3 1600 UT 08 September 2015

Figure 5 shows results for 1600 UT 08 September 2015 in the same format. This was during the period of near 0 nT SYM/H between the two storm minima. It was during a time period when both  $B_z$  and  $B_y$  are positive (See Figure 1.). Again the radial location of the partial pressure peaks are similar. The TWINS peak, however, has moved to the noon/dusk sector. It has continued to move westward from it positions in Figures 3 and 4. This could be the classic drift due to magnetic field gradient and curvature as originally observed in IMAGE/HENA ENA images by Brandt et al., (2001). In contrast to the TWINS pressure profile, the CIMI pressures reflect a nearly symmetric ring current. While ASY/H was relatively low at this time, it did show a small peak (See Figure 1.). Both the CIMI/RCM and the CIMI/Weimer 2K results show a region of parallel pressure anisotropy at large radii that almost circles the Earth. The TWINS

results show only perpendicular pressure anisotropy.

# 5.2.4 0200 UT 09 September 2015

Figure 6 shows results for 0200 UT 09 September 2015 in the same format. This is early in the main phase of the second minimum in SYM/H (See Figure 1.). The TWINS equatorial ion partial pressure peak is at a larger radius and in the midnight/dawn sector in contrast to the CIMI results where the peaks are in the dusk/midnight sector. There is considerably more spatial structure in the TWINS results. The strongest TWINS peak extends well into the dusk/midnight sector with a region near the same location as the CIMI peaks and with another at a larger radius in the dusk/midnight sector. There is an even larger difference in the pressure anisotropy. The parallel region at large radii in the CIMI result is even more parallel but is again absent in the TWINS result. The small intense parallel region at very small radius in the TWINS plot is a region of very low flux and therefore not a reliable ratio. At this time, the AE index was rising sharply as was the ASY/H index (See Figure 1.).

## 5.2.5 0400 UT 09 September 2015

Figure 7 shows results for 0400 UT 09 September 2015 in the same format. This was just 2 hours later than the time shown in Figure 6. It was near the end of the main phase of the second minimum in SYM/H (See Figure 1.). Again the TWINS peak is in the midnight/dawn region whereas the CIMI peaks appear in the dusk/midnight region, but the radial location is very nearly the same. This time, however, the TWINS peak extends past dawn and not into the pre-midnight region. Even though the MLT location of the CIMI/RCM and the CIMI/Weimer 2K peaks are nearly the same, the CIMI/Weimer 2K maximum extends to almost noon. The pressure anisotropy shows features very similar to those seen 2 hours previously (See Figure 6.) .The AE index has been at fairly high values for about an hour and the ASY/H index is beginning to rise sharply again (See Figure 1.).

## 5.2.6 1800 UT 09 September 2015

Figure 8 shows results from 1800 UT 09 September 2015 in the same format. At this time SYM/H (See Figure 1.) shows that the second storm was a few hours into a slow recovery. There are 4 distinct peaks in the TWINS equatorial ion partial pressure distribution. The highest is at large radius, about 7  $R_E$ , in the dusk/midnight sector. There is another lower peak, also at large radius in the noon/dusk sector. There are two peaks at a similar radius as the CIMI peaks. This interval is an example of multiple peaks in the ring current that have been inferred from insitu measurements (Liu et al., 1987), and seen in analysis of ENA images (Perez et al., 2015). The parallel pressure anisotropy in the CIMI results is again present, but it is smaller and weaker than at previous times. Again TWINS does not show this feature.

## 5.2.7 1700 UT 10 September 2015

Figure 9 shows results from 1700 UT 10 September 2015 in the same format. At this time the second storm was well into its slow recovery, SYM/H was beginning a small dip, there was a peak in the AE index, and ASY/H had a weak peak. (See Figure 1.) The partial pressure profiles for CIMI/RCM and CIMI/Weimer 2K are symmetrical with a peak in the dusk/midnight sector. The TWINS partial pressure peak is closer to dusk. This interval is in contrast to results at earlier times in the storm. The TWINS partial pressure peak is at a larger radius, and there is very little flux in the dawn/noon sector. The CIMI pressure anisotropies again show a region of strong parallel pitch angles that is not seen in TWINS.

#### **6** Discussion

Injections from the plasma sheet are thought to be the primary source of ring current protons in the inner magnetosphere, i.e., those that are observed by TWINS. Electric and magnetic fields determine the ultimate path of the injected ions, i.e., whether they reach locations close enough to the Earth where the magnetic gradient and curvature drifts are strong enough to exceed the electric drift forming the ring current or whether they drift out to the magnetopause. The locations of the partial pressure peaks from the CIMI/RCM and the CIMI/Weimer 2K simulations and the TWINS observations during the 4-day period, 07-10 September 2015, show that the peaks are usually in the dusk/midnight sector. (See Figure 2b) This phenomenon is consistent with analysis of data at geosynchronous orbit (Birn et al., 1997). Nevertheless the TWINS observations show partial pressure peaks that are often at larger radii than the CIMI simulations, even when they are in the dusk/midnight sector (See Figure 2a.). The fact that the CIMI/Weimer peaks are generally closer to dusk than the CIMI/RCM. (See Figure 2b.) is consistent with simulations reported by Fok, et al. (2003). The TWINS MLT locations are closer to midnight and in the midnight /dawn sector more frequently than the CIMI results. This suggests that there are often enhanced electric shielding and effects from localized and short time injections that are not present in the CIMI simulations.

To understand how the electric shielding works to affect the paths of the injected particles, we note that the convection electric field from the solar wind is mapped into the magnetosphere along open field lines into the polar ionosphere. It is then shielded from penetrating to lower latitudes and therefore further into the inner magnetosphere by the Birkeland region 2 currents driven by pressure gradients in the ring current. During geomagnetic storms when there is a sharp turn in the z-component of the interplanetary magnetic field (IMF) from negative to positive (See row 2 of Figure 1.), the accompanying electric field in the ionosphere associated with the Region 2 currents can produce what is referred to as over-shielding. See for example Jaggi and Wolf (1973). There are also neutral disturbance dynamo electric fields in the ionosphere that affect electric shielding. Localized and short time injections may contribute to the complexity of these effects.

Looking in detail reveals an even more complex story. Figures 3-9 show comparisons of the partial pressure profiles during different phases of the storms. In the main phase of the first storm (See Figure 3.), while there is a significant AE index and ASY/H asymmetry (See Figure

1.), the observed TWINS peak is at midnight while the simulated peaks are more toward dusk. During the rapid recovery phase of the first storm, (See Figure 4.) when the AE index is smaller (See Figure 1.), the observed and simulated partial pressure peaks are at approximately the same radius, and all are in the dusk/midnight sector. During the period between the two storms (See Figure 5.) when there is very little geomagnetic activity, i.e., SYM/H near 0 nT (See Figure 1.), the observed partial pressure peak has drifted more westward than the simulated peaks, even going past dusk (See Figure 5.). Another feature to note is the symmetry of the ring current in the CIMI simulations whereas the TWINS observations show a gap in the dawn/noon sector. The ASY/H index shows a small peak at this time (See Figure 1.) This suggests time dependence in the electric and magnetic fields that is not present in the CIMI simulations.

It is in the second storm (Figures 6-8) that the TWINS observations begin to show more spatial and temporal structure than the CIMI simulations. In Figure 6, early in the main phase, the TWINS observations show the main partial pressure peak near 6 R<sub>E</sub> and 3 MLT while the simulated peaks are near 4  $R_E$  and 20 MLT. But there is also a strong observed pressure region in the same area as the simulated peaks. Just 2 hours later, the simulated pressure shows little change, but the observed main peak extends farther eastward, and the relative pressure in the dusk/midnight region has weakened relative to the main peak. Fourteen hours later in the recovery phase of the second storm, the simulated peaks have not changed significantly, whereas the TWINS observed peaks are dramatically different (See Figure 8.).. There are 4 pressure peaks. The strongest peak is at 7  $R_E$  and just westward of midnight. At smaller radii, there is a weaker peak near the location of the simulated peaks as well as one on the dawn side past midnight. There is another weaker peak at large radius near noon. It should be noted that there is strong AE activity and that ASY/H has significant values during this period (See Figure 1.). This activity suggests that there may be variations in the electric and magnetic fields produced by spatial and time dependence of the location of the ion injections that are not present in the CIMI simulations.

The increased structure in the partial pressure distributions as observed by TWINS is especially dramatic during the recovery phase of the second storm. (See Figure 8.) There is strong AE activity and the largest values of ASY/H during this period. In the late recovery of the second storm (See Figure 9.), the CIMI simulations show a symmetric ring current as expected (Pollock et al., 2001). The TWINS results are not symmetric and have a peak at large radius in the dusk/midnight sector. There is some AE activity and a rise in the ASY/H index at this time.

Figures 3-9 also show comparisons of the pressure anisotropy during the different phases of the storm. The pressure anisotropies at the partial pressure peaks are generally in good agreement among the 3 results presented here, i.e., the pitch angle distributions are more perpendicular than parallel. The CIMI simulations, however, show a consistent region of parallel anisotropy at radii outside the pressure peak. The degree to which the pitch angle distributions are more parallel increases until the early recovery phase of the second storm (See Figure 8.) where it weakens but then strengthens again in the late recovery phase. This feature is seen by TWINS only in the main phase of the first storm (See Figure 3.) and perhaps very faintly in the early recovery phase of the are injected at the boundary of the CIMI simulations, located at 10  $R_E$  for those shown here, have an isotropic pitch

angle distribution. As they are accelerated while conserving the first adiabatic invariant to enter the region observed by TWINS, i.e. an outer radius of 8  $R_E$ , their pitch angle distributions become parallel because the energy increase exceeds what can be absorbed in the perpendicular pitch angles while still conserving the first adiabatic invariant. One mechanism for reducing the parallel anisotropy is wave-particle interactions which are not included in the CIMI simulations..

Another possible contributing factor to the differences between the observations and simulations is the input to the CIMI model used in these simulations. Following Fok et al.(2014), the ion distribution at the boundary of the CIMI simulations in this study is an isotropic, Maxwellian distribution at a radius of 10  $R_E$  at all MLT. The density and temperature of the Maxwellian is taken to have a linear relation to the solar wind density and solar wind velocity respectively (Borovsky et al., 1998; Ebihara and Ejiri, 2000). This produces a relatively smooth time variation in the input which has been shown to be successful in matching the general features of SYM/H (Buzulukova et al., 2010), but does not match the more rapid variations as a function of time. It has also been shown that varying the spatial dependence of the input along the boundary can have a significant effect on the location of the pressure peaks (Zheng et al., 2010). Likewise Buzulukova et al. (2010) showed that input of non-isotropic pitch angle distributions can affect the comparison between the CIMI simulations and the ENA observations.

There is significant experimental evidence for temporal and spatial variations in the injection of ions into the trapped particle region of the ring current (e.g., Birn et al., 1997; Daglis et al., 2000; Lui et al., 2004). Bursty bulk flows associated with near-Earth magnetic reconnection events have been frequently observed in the magnetotail (Angelopoulos et al., 1992). These fast flows have been observed to have a 1-3  $R_E$  width in the dawn-dusk direction (e.g., Angelopoulos et al., 1996; Nakamura et al., 2001; Angelopoulos et al., 2002). Magnetic flux ropes flowing Earthward have also been observed (e.g., Slavin et al., 2003; Eastwood et al., 2005; Imber et al., 2011). Short time, spatially limited injections into the inner magnetosphere have also been seen in 3D hybrid simulations. (e.g. see Lin et al., 2014.) Thus it is reasonable to suppose that the additional spatial and temporal structure in the partial pressure profiles observed during this storm is due to effects not yet incorporated into the simulations.

Buzulukova et al. (2008) combined the Comprehensive Ring Current Model (CRCM) (Fok et al., 2001) and the Dynamical Global Core Plasma Model (Ober et al., 1997) to model features of the plasma sphere observed by the Extreme UltraViolet (EUV) instrument on the Imager for Magnetosphere-to-Aurora Global Exploration (IMAGE) (Burch, 2000) on 17 April 2002. They found that injections from the plasma sheet that were localized in magnetic local time (MLT) explained observed undulations of the plasmasphere. Some features of an inductive electric field were included through the use of a time dependent magnetic Tsy96 (Tsyganenko and Stern, 1996) magnetic field model.

Likewise, Ebihara et al. (2009) compared CRCM simulations with midlatitude Super Dual Auroal Radar Network (SuperDARN) Hokkaido radar observations of fluctuating iononspheric flows on 15 December 2006. Using input from geosynchronous satellites to model the temporal and spatial variations of the plasma sheet input to the inner magnetosphere, they were able to show that the resulting pressure variations in the ring current were responsible for field aligned currents and matched the dynamics of the observed subauroral flows. The results from the CRCM also showed multiple pressure peaks inside of 4  $R_E$ . This is indicative of a strong connection between the dynamics of the ring current pressure distribution and the rapid temporal characteristics of the subauroral plasma flow during a geomagnetic storm.

The comparisons between the observations and the simulations presented here give a view not available from in-situ measurements. To further elucidate this phenomenon, we present in Figure 10 the paths of particles injected into the inner magnetosphere calculated using the CIMI simulations that provide additional support for concluding that the observations may show effects from enhanced electric shielding and localized and short time injections. The focus is upon the time 1800 UT on 9 September 2015 during the second storm. As shown in Figure 8, the TWINS observations show multiple peaks in contrast to the single peak in the CIMI simulations. For each of the 4 partial pressure peaks observed by TWINS, we show the energy spectrum (left column) and the paths of particles that reach the location of the pressure peaks (right column). The energy spectra show two energy maxima, one below 20 keV and the largest maxima above 40 keV. The ion paths are calculated with the CIMI model using the RCM fields. The path shown is of a particle with an energy of 46 keV when it reaches the respective pressure peaks, i.e., the energy at the maximum of the energy spectra shown in the left hand column. The TWINS partial pressure configuration from Figure 8 is repeated in gray scale so as to highlight the paths. In each case the pressure peak is shown by a black square. Along the path there are stars every 10 minutes. The color of the stars indicate the ion energy as it moves along its path. (See color bar.)

For Peak 1, the 46 keV particle enters at 10 R<sub>E</sub> in the midnight/dawn sector. The time from injection to reaching this peak in the outer magnetosphere is approximately 20 minutes. For Peak 2, which is at a smaller radius, a 46 keV ions arrives at the peak from the dawn/midnight sector after approximately 2 <sup>1</sup>/<sub>2</sub> hours. This peak observed by TWINS is very near the pressure peak that appears in the CIMI simulations. (See Figure 8.) Peak 3 is at a similar radius as Peak 2, but it is on the dawn side of midnight. The path of a 46 keV particle followed backwards in time from this peak location does not show an injection location after completing nearly 3 orbits of the Earth in approximately 12 hours. This partial pressure peak observed by TWINS may not be consistent with the RCM fields in the CIMI model. Peak 4 is in the noon/dusk sector. A 46 keV particle reaches this peak after approximately 3 <sup>3</sup>/<sub>4</sub> hours and 1 orbit of the Earth. It enters the inner magnetosphere in the same sector, i.e., the midnight/dawn sector, as the particle that reached the location of Peak 1, but it was injected much earlier. The different locations and times of the entrance of the ions at the peaks of the energy spectra of the 4 pressure peaks 1, 2, and 4 observed by TWINS at 1808 UT on 9 September 2015 suggest spatial and temporal variations in the injections from the plasma sheet. The fact that the calculated path for Peak 3 does not show an injection may indicate variations in the fields not captured in the models.

#### 7 Summary and Conclusions

We have presented, for the first time, direct comparisons of the equatorial ion partial pressure distributions and pitch angle anisotropy obtained from TWINS ENA images and CIMI

simulations using both an empirical Weimer 2K and the self-consistent RCM electric potentials for a 4-day period, 7-10 September 2015. There were two moderate storms in succession during this period (See Figure 1.). In most cases, we find that the comparison of the general features of the ring current in the inner magnetosphere obtained from the observations and simulations are in agreement. Nevertheless, we do see consistent indications effects of enhanced electric shielding and localized and short time injections from the plasma sheet in the observations. The simulated partial pressure peaks are often inside the measured peaks and are more toward dusk than the measured values (See Figure 2.). There are also cases in which the measured equatorial ion partial pressure distribution shows multiple peaks that are not seen in the simulations (See Figure 8.). This occurs during a period of intense AE index. The observations suggest time and spatially dependent injections from the plasma sheet that are not included in the simulations. The paths of the ions that enter the inner magnetosphere calculated with the CIMI model using the self-consistent RCM fields support this interpretation.

The simulations consistently show regions of parallel anisotropy spanning the night side between approximately 6 and 8  $R_E$  (See Figures 3-9.). This is thought to be a result of the increasing energy of the particles as they come enter the simulation region at 10 RE with isotropic pitch angle distributions. The particles are entering regions of stronger magnetic field so conservation of the first adiabatic invariant requires the perpendicular velocity to increase, but it is not adequate to accommodate the increase in energy. So the parallel velocity must increase. Nevertheless the parallel anisotropy is seen in the observations only during the main phase of the first storm. Localized and short time injections may produce ions that are injected with perpendicular pitch angle distributions that would result in the observed nearly isotropic pressure anisotropy.

*Acknowledgments.* OMNI solar wind data are accessible via CDAWeb at <u>https://cdaweb.gsfc.nasa.gov/.</u> TWINS data are accessible to the public at <u>http://twins.swri.edu.</u> Geomagnetic activity indices are also available from the World Data Center for Geomagnetism in Kyoto, ttps://wdc.kugi.kyoto-u.ac.jp/wdc/Sec3.html.

This work was supported by the TWINS mission, a part of NASA's Explorer program. We thank the World Data Center for Geomagnetism, Kyoto for suppling Real Time Dst and AE indices. We also thank the ACE and Wind plasma and magnetometer teams for L1 data and the OMNI data set for their propagation of these data.

Significant parts of the calculations in this study were performed on the Auburn University High Performance and Parallel Computing Facility.

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

# **Figure Captions**

**Figure 1**. The solar wind parameters and geomagnetic indices for the two storms during the period 07-10 September 2015. The data is from the OMNI data base (https://omniweb.gsfc.nasa.gov/html/omni\_min\_data.html).

**Figure 2**. Plot of the ion equatorial pressure peak as a function of time during the 4-day period 07-10 September 2015. (a) the radial location and (b) the MLT location. The green triangles mark the locations obtained from the TWINS ENA images, the red line from the CIMI/Weimer simulations and the orange line from the CIMI/RCM simulations.

**Figure** 3. The ion equatorial pressure (first row) and pressure anisotropy (second row) for 2200 UT 07 September 2015 from the CIMI/RCM simulations (first column), from the TWINS ENA images (second column), and the CIMI/Weimer simulations (third column). The stars mark the location of the peaks.

**Figure** 4. The ion equatorial pressure and pressure anisotropy for 0400 UT 08 September 2015 in the same format as Figure 3.

**Figure 5**. The ion equatorial pressure and pressure anisotropy for 1600 UT 08 September 2015 in the same format as Figure 3.

**Figure 6**. The ion equatorial pressure and pressure anisotropy for 0200 UT 09 September 2015 in the same format as Figure 3.

**Figure 7**. The ion equatorial pressure and pressure anisotropy for 0400 UT 09 September 2015 in the same format as Figure 3.

**Figure 8**. The ion equatorial pressure and pressure anisotropy for 1800 UT 09 September 2015 in the same format as Figure 3.

**Figure 9**. The ion equatorial pressure and pressure anisotropy for 1700 UT 10 September 2015 in the same format as Figure 3.

**Figure 10**. Paths of 46 keV particles, the energy of protons at the maximum flux (See left column.) that reach the 4 pressure peaks observed by TWINS as shown in Figure 8. The observed pressure is shown in grey scale. The locations of the peaks are shown by black squares. The energy of the particle is indicated by the color of the stars that are spaced 10 minutes apart. The units of the color bars are keV. The energies span the range of the particle energies along their paths.