# Peer review of "Dynamics of a Geomagnetic Storm on 7-10 September 2015 as Observed by TWINS and Simulated by CIMI"

_Annales Geophysicae, 2018_

## Referee Comment (RC1) · Anonymous Referee #1 · 11 Jul 2018

[GENERAL COMMENTS]

This paper presents the equatorial plasma pressure distributions obtained by the TWINS observation and by the drift kinetic simulation CIMI for the moderate storms of 7-10 September 2015. The general features of the plasma pressure in the inner magnetosphere are similar to each other, whereas some differences are found in terms of peak location, anisotropy, and spatial distribution. The authors attributed the differences to the shielding effect and spatially-localized, short-duration injections of hot plasma.

The direct comparison between a sophisticated observation and an advanced drift ki-

netic equation is highly valuable, and is promising to overcome the difficulties arising from in-situ satellite observations. The provided data is basically very interesting, and I admire the authors' efforts to derive the pressure and anisotropy. However, I have 3 major concerns as follows. First, the physical interpretations made by the authors are unclear. Because of the lack of proper interpretations, I cannot catch new scientific knowledge, or insights in the current version of the manuscript. Secondly, the reliability of the plasma pressure obtained by TWINS is also unclear. The spectral shape of the ion flux is almost the same at 4 different points, which seems unlikely to occur. Thirdly, the plasma pressure mentioned in this paper is "partial" so that the "true" distribution of the plasma pressure would be different. Careful description is needed when the authors intend to say the distribution of the pressure.

[SPECIFIC COMMENTS]

(1) On the interpretations. The authors concluded that the difference between the observation and the simulation can be best explained by enhanced electric and magnetic shielding and/or spatially-localized, short-duration injections. First of all, please explain the meaning of the electric and magnetic shielding in more detail. Most of the readers may not understand the meaning of it. The electric shielding is supposed to result from the ionospheric electric field redistributed by the Region 2 field-aligned current. What is the magnetic shielding? What is the expected effect of the shielding on the pressure distribution and pressure anisotropy? CIMI/RCM takes into account the shielding. What physical processes or parameters does CIMI/RCM need to consider properly to explain the observations? Have the authors tested CIMI/RCM with different conditions/parameters to explain the observations? Secondly, please explain the expected effect of spatially-localized, short-duration injections on the pressure and anisotropy. Have the authors modeled spatially-localized, short-duration injections to explain the observations? Thirdly, please explain the reason why the CIMI result always shows parallel anisotropy of the plasma pressure in the dawn-midnight-dusk region. The pressure anisotropy is largely different from the observations. Detailed explanation is needed.

(2) On reliability of the plasma pressure. In Figure 10, the differential fluxes of the ions are shown as a function of energy at 4 points. The intensity of the flux is different, but the spectral shape is almost the same with each other. Why is the spectral shape of the flux almost the same at the 4 points? According to in-situ observations, the spectral shape of the flux depends on L-value and magnetic local time (e.g., Milillo et al., 2001, 10.1029/2000JA900158), so that it seems kquite unlikely to be the same spectral shape at 4 points. Please explain the validity of the spectral shape of the flux and the plasma pressure distribution presented in this paper.

(3) On the plasma pressure. I suppose that the plasma pressure was calculated from integration of the differential flux over the energy range from 2.5 keV to 97.5 keV. The energy range is probably insufficient to cover all the ions trapped in the inner magnetosphere because the ions with energy greater than 100 keV is also known to contribute to the plasma pressure (energy density) largely (e.g., Smith and Hoffman, 1973, 10.1029/JA078i022p04731; Williams, 1983, 10.1016/0032-0633(81)90124-0). If the high energy ions remained during these storms, there would be another peak of the pressure, which may stay at L $\sim$ 2.5 – 3.0. I recommend discussing possible impacts of the high energy ions (>100 keV) on the conclusion. I also recommend emphasizing that the plasma pressure distribution is "partial" so that the pressure distribution is incomplete.

[MINOR COMMENTS]

Introduction: I recommend citing papers related to plasma pressure distribution and anisotropy observed by satellites, for example, De Michelis et al. (1999, 10.1029/1999JA900310), Ebihara et al. (2002, 10.1029/2002GL015430), and Lui (2003, 10.1029/2003GL017596).

Line 47-57: Simulation results with different electric field and/or magnetic field models have been conducted by Angelopoulos et al. (2002, 10.1029/2001JA900174) and Ebihara et al. (2004, 10.5194/angeo-22-1297-2004).

Line 58-63: This paragraph seems not to provide information. What key spatial features do Wang et al. (2011) find?

Line 311: The equatorial pressure p_eq is difficult to understand. Please explain how to derive p_eq.

Line 489-496: Ebihara et al. (2009) also showed multiple peaks of the plasma pressure distribution in the inner magnetosphere by introducing temporal changes in the distribution function at the outer boundary of CRCM. It would be worth mentioning that the rapid changes in the distribution function in the plasma sheet could result in the multiple peaks of the plasma pressure.

Line 499 "But they do not provide incontrovertible evidence for the effects of spatially and temporally dependent injections into the inner magnetosphere." This sentence is difficult to understand.

Figure 10, caption: Please indicate the unit of the color bar (probably in keV), and pitch angle of the particle. What is the meaning of "Minimum – Maximum Energy for Each Path"?

Line 520, "Peak 5" Does it mean "Peak 4"?

Line 497-527: The spectral shape of the differential flux of the ions is almost the same at the 4 points. Please explain the validity of the differential flux derived from TWINS? At Peak 3, the ion is inaccessible from the outer boundary. I recommend tracing the ion trajectory backward in time by starting at slightly different points.

Line 546-548: "This is not unexpected as the ions are being injected into regions of higher magnetic field, and conservation of the first adiabatic invariant would predict the enhancement of parallel pitch angles." I cannot understand this meaning. Please explain the reason why the conservation of the first adiabatic invariant results in the pressure anisotropy dominated by the parallel component?

Line 548-550: "Nevertheless the parallel anisotropy is seen in the observations only

during the main phase of the first storm. This is also an indication of stronger electric and magnetic shielding." Please explain the reason why the stronger shielding results in the parallel anisotropy?

———————————————

---

## Author Comment (AC1) · 6 Aug 2018

[GENERAL COMMENTS]

This paper presents the equatorial plasma pressure distributions obtained by the TWINS observation and by the drift kinetic simulation CIMI for the moderate storms of 7-10 September 2015. The general features of the plasma pressure in the inner magnetosphere are similar to each other, whereas some differences are found in terms of peak location, anisotropy, and spatial distribution. The authors attributed the differences to the shielding effect and spatially-localized, short-duration injections of hot plasma.

The direct comparison between a sophisticated observation and an advanced drift ki-

netic equation is highly valuable, and is promising to overcome the difficulties arising from in-situ satellite observations. The provided data is basically very interesting, and

I admire the authors' efforts to derive the pressure and anisotropy.

**The authors thank the referee for the thoughtful and helpful comments. We respond positively below to each question and comment individually. Nevertheless, we cannot answer every question posed by referee with a full, unambiguous explanation. Such requires extensive investigations that are underway but are beyond the scope of this paper. We believe what many have said, i.e., good research raises at least as many questions as it answers. Our responses are shown in bold font for ease of distinguishing our responses and the referee's comments.**

However, I have 3 major concerns as follows. First, the physical interpretations made by the authors are unclear. Because of the lack of proper interpretations, I cannot catch new scientific knowledge, or insights in the current version of the manuscript.

**We certainly hope that our responses are sufficient so that the referee and other readers are able to "catch new scientific knowledge or insights. . . "**

Secondly, the reliability of the plasma pressure obtained by TWINS is also unclear. The spectral shape of the ion flux is almost the same at 4 different points, which seems unlikely to occur.

**The intent of showing the spectral shapes in Figure 10 was not to present an extended discussion of the spectra but rather to indicate that were two peaks, one at low energy, i.e., below 20 keV, and a second near 40 keV and to illustrate why the paths of 40 keV particles are shown. We are not sure that this is unlikely to occur. Details of the magnitude and shape of the energy spectra will be addressed in an extensive study that is underway.**

**As to the overall reliability of the plasma pressure, that is a difficult question to answer definitively and quantitatively. In the description of the methodology used to obtain the ion pressure from the TWINS ENA images, we have referenced the extensive testing of the ion distributions obtained from the TWINS ENA images. (See Section 2.2, lines 190 ff in the version of the paper where corrections are accepted and lines 194 ff in the version of the paper where corrections are marked.)**

Thirdly, the plasma pressure mentioned in this paper is "partial" so that the "true" distribution of the plasma pressure would be different. Careful description is needed when the authors intend to say the distribution of the pressure.

**The referee is correct. What we show in this paper are "partial" pressures. The intent is to have the TWINS and CIMI results consider as nearly as possible a similar energy range. In the revised paper, all references to the particular pressures from TWINS and CIMI are now designated as partial pressure.**

[SPECIFIC COMMENTS]

(1) On the interpretations. The authors concluded that the difference between the ob- servation and the simulation can be best explained by enhanced electric and magnetic shielding and/or spatially-localized, short-duration injections. First of all, please explain the meaning of the electric and magnetic shielding in more detail. Most of the readers may not understand the meaning of it. The electric shielding is supposed to result from the ionospheric electric field redistributed by the Region 2 field-aligned current.

**Excellent suggestion. We include in the revised version the following explanation as to what we mean by electric shielding. (See Section 6, lines 436 ff in the version of the paper where corrections are accepted and lines 442 ff in the version of the paper where corrections are marked.)**

What is the magnetic shielding?

**"magnetic shielding" was an improper term. Better to say enhanced electric field shielding and/or induction electric fields caused by spatially-localized, short duration injections. The term is omitted in the revised version.**

What is the expected effect of the shielding on the pressure distribution and pressure anisotropy?

**It has been demonstrated that the electric shielding can affect the ring current morphology. (See Section 1, lines 77 ff in the version of the paper where corrections are accepted and lines 80 ff in the version of the paper where corrections are marked.) The purpose of this paper is to show much more explicitly what are the expected effects. We are currently undertaking a project to couple the CIMI code in the inner magnetosphere with a 3D hybrid code that simulates the rest of the magnetosphere. We expect the results to address these issues in even greater detail.**

CIMI/RCM takes into account the shielding. What physical processes or parameters does CIMI/RCM need to consider properly to explain the observations?

**The CIMI simulations presented in this paper do not have localized injections into the inner magnetosphere. This has been demonstrated to have an effect on the pressure distributions. (See Section 6, lines 520 ff in the version of the paper where corrections are accepted and lines 530 ff in the version of the paper where corrections are marked.) While the CIMI simulations in this paper do include some contributions from induction electric fields, the Tsyganenko magnetic field is not updated on short enough time scale to capture all of the induction electric fields. We, of course, do not know whether the steps we are taking to provide localized and short term injection effects will answer all the questions.**

Have the authors tested CIMI/RCM with different conditions/parameters to explain the observations?

**There have been investigations that demonstrate that changing the input at the boundary of the CIMI simulations does impact the ring current morphology. Also it has been shown that injecting non-isotropic pitch angle distributions impacts the parameter of the ring current. (See Section 6, lines 509 ff in the version of the paper where corrections are accepted and lines 516 ff in the version of the paper where corrections are marked.) The authors have not tried, however, to reverse engineer the input to the CIMI simulations presented in this paper to attempt to match the data.**

Secondly, please explain the expected effect of spatially-localized, short-duration injections on the pressure and anisotropy.

**As described above and is illustrated to some extent in previously referenced work, the authors expect that spatially-localized, short-duration injections will impact the spatial and temporal locations of the pressure peaks. We also expect it to be a key factor in explaining the observation of multiple peaks in the ring current.**

Have the authors modeled spatially-localized, short-duration injections to explain the observations?

**As stated above, it has been demonstrated that spatially localized injections can affect the ring current morphology, but we have not tried to match the observations for this particular storm without some experimental or theoretical guidance. There is an ongoing investigation to couple CIMI with a 3D hybrid simulation of the injections from the tail explicitly intended to address this question.**

Thirdly, please explain the reason why the CIMI result always shows parallel anisotropy of the plasma pressure in the dawn-midnight-dusk region. The pressure anisotropy is largely different from the observations. Detailed explanation is needed.

**An explanation is given in the revised paper. (See Section 6, lines 490 ff in the version of the paper where corrections are accepted and lines 495 ff in the version of the paper where corrections are marked.) Whether this is the complete explanation is uncertain at this time. The authors expect to develop a more definitive explanation as part of an ongoing investigation to couple CIMI with a 3D hybrid simulation of the injections from the tail.**

(2) On reliability of the plasma pressure. In Figure 10, the differential fluxes of the ions are shown as a function of energy at 4 points. The intensity of the flux is different but the spectral shape is almost the same with each other. Why is the spectral shape of the flux almost the same at the 4 points? According to in-situ observations, the spectral shape of the flux depends on L-value and magnetic local time (e.g., Milillo et al., 2001, 10.1029/2000JA900158), so that it seems kquite unlikely to be the same spectral shape at 4 points. Please explain the validity of the spectral shape of the flux and the plasma pressure distribution presented in this paper.

**We reference the published examples of validation of all the features of the ion distributions through comparisons with in-situ measurements. (See Section 2.2, lines 190 ff in the version of the paper where corrections are accepted and lines 194 ff in the version of the paper where corrections are marked.) These examples were chosen because the satellites happen to be in the right place at the right time to see the features of interest. It is true, of course, that such comparisons do not guarantee the complete validity of the current results.**

**The main reason for showing the measured energy spectra was to show why the paths of 46 keV ions were used to display the location and time of the injections of the ions that had 46 keV energy at the peaks. What the authors think has the relevant validity in this case is the high energy a low energy peaks in the spectra.**

**The outstanding work by Millilo et al , 2001 will make an important contribution to our current investigation into the details of the energy spectra during geomagnetic storms during the TWINS mission. The fact that the average results presented in their paper is for AE < 100 nT does not directly invalidate the results presented here.**

**Finally, as was stated above, the focus of this paper is not the details of the energy spectra. If the referee would prefer, we could remove them from Figure 10.**

(3) On the plasma pressure. I suppose that the plasma pressure was calculated from integration of the differential flux over the energy range from 2.5 keV to 97.5 keV. The energy range is probably insufficient to cover all the ions trapped in the inner magnetosphere because the ions with energy greater than 100 keV is also known to contribute to the plasma pressure (energy density) largely (e.g., Smith and Hoffman, 1973, 10.1029/JA078i022p04731; Williams, 1983, 10.1016/0032-0633(81)90124-0). If the high energy ions remained during these storms, there would be another peak of the pressure, which may stay at L 2.5 – 3.0. I recommend discussing possible impacts of the high energy ions (>100 keV) on the conclusion. I also recommend emphasiz- ing that the plasma pressure distribution is "partial" so that the pressure distribution is incomplete.

**The referee is correct, the plasma pressure presented in this paper should be referred to as partial pressure because it was calculated by integrating from 2.5 to 97.5 keV. The paper by Smith and Hoffman, 1973 certainly shows that higher energies can make significant contributions to the energy density (pressure). It is to be noted, however, that they say**

**"To contrast the development of the ring current for the two storms, we now consider those protons (1- to 138-kev protons were used) which contribute substantially to the storm-time ring current. While protons in this energy regime contribute only 20% or less to the total energy density out to L -• 4 during magnetically quiet periods, their enhancement during magnetic storms, combined with a depletion of protons with energies greater than about 170 kev, make them the dominant (greater than 90%) contributors to the storm-time energy densities."**

**The paper is referenced. (See Section 2.2, line 179 in the version of the paper where corrections are accepted and line 183 in the version of the paper where corrections are marked.)**

The paper by William, 1983, describes the state of observations at that time with the conclusion, "It is found that the ring current energy density composition still has not been observed."

The authors wholeheartedly agree with the recommendations of the referee.

[MINOR COMMENTS]

Introduction: I recommend citing papers related to plasma pressure distribution and anisotropy observed by satellites, for example, De Michelis et al. (1999, 10.1029/1999JA900310), Ebihara et al. (2002, 10.1029/2002GL015430), and Lui (2003, 10.1029/2003GL017596).

**Definitely. The authors apologize for not recognizing these papers. (See Section 1, line 65, 70, 72 , in the version of the paper where corrections are accepted and line 68, 73, 75 in the version of the paper where corrections are marked.)**

Line 47-57: Simulation results with different electric field and/or magnetic field models have been conducted by Angelopoulos et al. (2002, 10.1029/2001JA900174) and Ebihara et al. (2004, 10.5194/angeo-22-1297-2004).

**Most definitely. The authors thank the referee for pointing us the these papers. (See Section 1, line 55, 61 in the version of the paper where corrections are accepted and line 58, 64, in the version of the paper where corrections are marked.)**

[Figure]

Line 58-63: This paragraph seems not to provide information. What key spatial features do Wang et al. (2011) find?

**The authors think it is somewhat harsh to say that the paragraph does not provide any information. In particular reference to the Wang et al. (2011) paper, the paper presents extensive data and comparisons with RCM modeling, but they are based upon statistical averages of events. The result from Wang et al (2011) that is relevant to this paper is what is stated in the last sentence of this paragraph. It is based upon the last sentence of the Abstract and the first sentence of the Summary.**

Line 311: The equatorial pressure p_eq is difficult to understand. Please explain how to derive p_eq.

**The authors are not sure why p_eq is difficult to understand. It is pressure at the equator as a function of position and pitch angle. It is the standard definition of pressure, i.e., the energy density of the ions, the integral of the distribution function times the velocity squared.**

Line 489-496: Ebihara et al. (2009) also showed multiple peaks of the plasma pressure distribution in the inner magnetosphere by introducing temporal changes in the distribution function at the outer boundary of CRCM. It would be worth mentioning that the rapid changes in the distribution function in the plasma sheet could result in the multiple peaks of the plasma pressure.

**A sentence has been added pointing out that the model calculations did show multiple pressure peaks inside of 4 $R_E$. (See Section 6, lines 534ff in the version of the paper where corrections are accepted and lines 545 ff, in the version of the paper where corrections are marked.)**

Line 499 "But they do not provide incontrovertible evidence for the effects of spatially and temporally dependent injections into the inner magnetosphere." This sentence is difficult to understand.

**The sentence has been removed.**

Figure 10, caption: Please indicate the unit of the color bar (probably in keV), and pitch angle of the particle. What is the meaning of "Minimum – Maximum Energy for Each Path"?

**An explanation has been added. (See Figure 10 caption, lines 534ff in the version of the paper where corrections are accepted and lines 1046 ff, in the version of the paper where corrections are marked.)**

Line 520, "Peak 5" Does it mean "Peak 4"?

**Yes. It has been corrected.**

Line 497-527: The spectral shape of the differential flux of the ions is almost the same at the 4 points. Please explain the validity of the differential flux derived from TWINS?

**I think we have addressed this issue in response to previous comments. The main reason for presenting these spectra is to motivate showing the paths of the 46 keV ions. As stated above, if the referee prefers, they can be removed.**

At Peak 3, the ion is inaccessible from the outer boundary. I recommend tracing the ion trajectory backward in time by starting at slightly different points.

**The authors are uncertain as to why the referee thinks the paths would show a chaotic dependence on the starting point. We tried a few slightly different points, and the result was the same.**

Line 546-548: "This is not unexpected as the ions are being injected into regions of higher magnetic field, and conservation of the first adiabatic invariant would predict the enhancement of parallel pitch angles." I cannot understand this meaning. Please explain the reason why the conservation of the first adiabatic invariant results in the pressure anisotropy dominated by the parallel component?

**A full explanation has been added. (See Section 6, lines 590ff in the version of the paper where corrections are accepted and lines 601 ff, in the version of the paper where corrections are marked.)**

Line 548-550: "Nevertheless the parallel anisotropy is seen in the observations only

during the main phase of the first storm. This is also an indication of stronger electric and magnetic shielding." Please explain the reason why the stronger shielding results in the parallel anisotropy?

**The statement has been changed to be consistent with responses to previous comments by the referee. (See Section 6, lines 595ff in the version of the paper where corrections are accepted and lines 609 ff, in the version of the paper where corrections are marked.)**
* * *

---

## Referee Comment (RC2) · Anonymous Referee #1 · 13 Aug 2018

The authors answered my comments properly except for one thing on the equatorial pressure (Line 311 in the first version of the manuscript). The authors stated that $p_{eq}$ is pressure at the equator as a function of position and pitch angle, and that it is the standard definition of pressure. If it is the standard definition of pressure, please cite relevant reference. The reason why I am asking is that readers may be eager to know how the authors obtained the terms, $P_\perp$ and $P_\parallel$. Here, I assume that $P_\perp$ and $P_\parallel$ are the pressure tensor components in the perpendicular and parallel components, respectively. Lui et al. (1987, 10.1029/JA092iA07p07459) show the equations to calculate $P_\perp$ and $P_\parallel$ as a function of velocity $v$ and the velocity distribution function $f$ (Eqs. 2 and 3 in Lui et al., 1987). The velocity distribution function $f$ is associated with the differential

flux that is directly measurable. What is the relationship between $p_{eq}$ and the measured value (probably differential flux derived from the ENA observation)?

---

## Author Comment (AC2) · 14 Aug 2018

Reply to Referee's Comment:

The Referee is correct, the equation as presented is unclear. Somehow the full response to the original comment was lost. The paragraph now reads as

The pressure anisotropy shown in Figure 3 is defined as

$$A = \frac{P_\perp - P_\parallel}{P_\perp + P_\parallel}$$

with

$$P_\perp = \int_{-1}^{+1} p_{eq}(\alpha) \sin^2 \alpha \, d\cos\alpha \quad \& \quad P_\parallel = 2\int_{-1}^{+1} p_{eq}(\alpha) \cos^2 \alpha \, d\cos\alpha$$

where α is the pitch angle and $p_{eq}$ is the equatorial pressure as a function of location and pitch angle

which was obtained from the energy dependent number flux deconvolved from the TWINS ENA images,

i.e.,

$$p_{eq} = \frac{2\pi}{m} \int_0^\infty E \, f(E, n, \cos\alpha) \, dE$$

where $f(E, n, \cos\alpha)$ is the number of ions per unit area, energy, and steradian. This definition is

derived from Braginskii (1965) and is consistent with previous formulations, e.g., Lui et al. (1987).

---

## Referee Comment (RC3) · Anonymous Referee #1 · 15 Aug 2018

The authors now present the meaning of the equatorial pressure $p_{eq}$ as $p_{eq} = \frac{2\pi}{m} \int E f \, de$, where $m$ is mass, $E$ is energy, and $f$ is the number of ions per unit area, energy and steradian. First of all, I am unsure if $f$ corresponds to the so-called differential number flux that is the number of ions per unit area, energy, time and steradian. If so, I have further comment.

The definition of the perpendicular and parallel pressure ($P_\perp$ and $P_{||}$) is as follows. $P_\perp = \frac{1}{2} \int m v^2 \sin^2 \alpha F d\mathbf{v}$ and $P_{||} = \int m v^2 \cos^2 \alpha F d\mathbf{v}$, where $F$ is the velocity distribution function. These equations can be derived from the original definition of pressure (probably Braginskii (1965) provided, too). Lui

et al. (1987) also present these equations. The velocity distribution function is given by $F = \frac{m}{v^2}f$. Substituting this into above equations, I have $P_\perp = \int\int \pi\sqrt{2m}f\sqrt{E}\sin^2\alpha d(\cos\alpha)dE = \int\int \pi\sqrt{2m}f\sqrt{E}dE\sin^2\alpha d(\cos\alpha) = \int p_{eq}\sin^2\alpha d(\cos\alpha)$ and $P_{||} = \int\int 2\pi\sqrt{2m}f\sqrt{E}\cos^2\alpha d(\cos\alpha)dE = \int 2p_{eq}\cos^2\alpha d(\cos\alpha)$, where $p_{eq} = \int \pi\sqrt{2m}f\sqrt{E}dE$. It seems that the definition of $p_{eq}$ is different from the authors'. The same equation is found in Eqs (7) and (8) in De Michelis et al. (1997, doi:10.1029/96JA03743).

Maybe I misunderstand, but I would like to make it clear.

I suggest avoiding the term 'equatorial pressure' because this term is confusing and misleading. The above equations can be applied for everywhere, not restricted in the equatorial plane.

---

## Referee Comment (RC4) · Anonymous Referee #1 · 16 Aug 2018

The authors revised the equation for the pressure terms. However, the equation seems to be different from the equation given by De Michelis et al. (1997, http://doi.wiley.com/10.1029/96JA03743) by a factor of $\pi m$. I am curious to know the reason why the equation is different.

The definition of the plasma pressure is changed. Does this change have any impact on the result? I suppose that lower energy protons may have more impact on the pressure.

I recommend removing $n$ from $F(E, n, \cos\alpha)$ because $F$ is an arbitrary function and $n$ is an independent variable.

---

## Author Comment (AC3) · 16 Aug 2018

Reply to Referee's Comment:

The question seems to revolve around 2 areas of confusion.

First, the use of the subscript "eq" to indicate that it is the equatorial pressure. The purpose was to communicate that it is at the equator that we actually calculate the pressure anisotropy. We agree, however, that this may be confusing so we agree to leave it out.

Second the definition of the symbols $f$ and $F$ in the equations. We suggest that to try to avoid this confusion, we suggest the following:

The pressure anisotropy shown in Figure 3 is defined as

$$A = \frac{P_\perp - P_\parallel}{P_\perp + P_\parallel}$$

with

$$\begin{Bmatrix} P_\perp \\ P_\parallel \end{Bmatrix} = \int_{-1}^{+1} d\cos\alpha \begin{Bmatrix} \sin^2\alpha \\ 2\cos^2\alpha \end{Bmatrix} \left( \int_0^\infty dE \sqrt{\frac{2E}{m}} F(E,n,\cos\alpha) \right)$$

where $\alpha$ is the ion pitch angle, $E$ is the ion energy, $n$ is the ion density, $m$ is the ion mass and $F(E,n,\cos\alpha)$ is the number flux per unit area, energy, time, steradian. This definition is derived from Braginskii (1965) and is consistent with previous formulations, e.g., Lui et al. (1987).

---

## Author Comment (AC4) · 16 Aug 2018

Reply to Referee's comment 4:

We very much appreciate the comments that have been made by the Referee. They have certainly made our paper better. But at this time, it seems that we having trouble communicating. We are not sure how to respond. Please see comments below: (The Referee's comments are repeated in italics.)

*The authors revised the equation for the pressure terms. However, the equation seems to be different from the equation given by De Michelis et al. (1997, http://doi.wiley.com/10.1029/96JA03743) by a factor of πm. I am curious to know the reason why the equation is different.*

I assume that the referee is referring to Eqs. (7) and Eqs (8) in De Michelis et al. (1997). Let's look at the unit of those eqs. For the pressure to have the correct units of energy per unit volume, the units of the "differential flux intensity", J in Eqs. (7) and (8) must be 1/(vol * m * v). The reason for the different factors is that J is not the flux we have in our equation. Our flux, F(E,n,cosα), as it says in the text, has units of #ions/(energy*time*area*steradian). One can check the units of the equation we have in the paper and they come out to be pressure. In fact if one substitutes the proper equation for a Maxwellian distribution into the equation in the paper, i.e.,

$$F(E,n,\cos\alpha) = \frac{n}{\sqrt{2m}(\pi T)^{3/2}} E\ e^{-E/T}$$

and performs the integrals, the result is $nT$, precisely what one expects for a Maxwellian.

*The definition of the plasma pressure is changed.*

The definition of the plasma pressure has not changed. We have been discussing a general definition of pressure. That has not changed, just the way it is presented has been made clearer with the help of the Referee.

*Does this change have any impact on the result? I suppose that lower energy protons may have more impact on the pressure.*

We assume this is in reference to the previous comment regarding the change in the definition of the pressure. We assume that the referee is referring to the fact that the pressure we calculate and present as results is the partial pressure, i.e., it is integrated from 2.5 to 97.5 keV for TWINS and 1 to 133 keV for CIMI. The referee correctly requested that we distinguish the pressure calculated in this paper as the partial pressure and we have done so. There is no change that would impact the results in this paper.

*I recommend removing n from F(E,n,cosα) because F is an arbitrary function and n is an independent variable.*

This comment may somehow be at the heart of the miscommunication that we are having at this time. Yes, it is in some sense arbitrary, i.e., in the expression for the pressure, it is whatever it is in a particular physical situation. In what we are presenting, however, the F(E,n,cosα) is definitely not an arbitrary function. For the TWINS results, it is what is obtained from the ENA images. For CIMI, it is what is obtained from the simulations. As stated above it has units of #ions/(energy*time*area*steradian). We feel that it makes the most sense to express the pressure in terms of what it is obtained, i.e., from the measurements and simulations. It is then integrated as expressed in the formulas to obtain the

pressures we present.  We might also note that previous publications of TWINS and CIMI results have shown the  average of $F(E,n,cos\alpha)$ over pitch angles as a function of energy.

---

## Referee Comment (RC5) · Anonymous Referee #2 · 17 Aug 2018

This manuscript shows comparisons between models and observations of the pressure peaks in the inner magnetosphere during a storm event. The comparison reveals both consistency and significant differences between the observations and model predictions. The authors discussed the possible cause of the difference (i.e., the missing transient structures in the simulation). The results of this manuscript are important for future improvement of models. However, there a few points that I would suggest the authors to address before I recommend the manuscript for publication:

- Line 276: varies -> vary

- Line 399-400: The authors start the sentence with both electric and magnetic shield-

ing but only explain magnetic shielding (gradient curvature drifts) in the later half of the sentence. The electric shielding is caused by the closure of region 2 current through the ionosphere, which creates a Peterson current, and thus electric field at lower latitudes than the region 2 current. This electric field, when mapped to the inner magnetosphere, cancels the original cross-tail electric field, so particles cannot ExB drift closer to Earth (see, e.g., Jaggi and Wolf, 1973). The electric shielding is more effective for low-energy particles. I do not think it is very important for the energy range which the authors are interested in.

- Line 455, and Line 547-548: 'parallel pitch angle anisotropy . . . first adiabatic invariant as they enter the inner magnetosphere': The conservation of first adiabatic invariant says that when a particle moves to a stronger magnetic field, it will have more perpendicular energy. Thus, the perpendicular anisotropy should increase instead of the parallel.

- Line 512-Line 527: This paragraph makes a strange comparison. To find the origin of the multiple pressure peaks, the authors uses particle tracing in the model, which does not have the multiple pressure peaks. As the authors said, the reason why the model cannot reproduce the observed multiple peaks is that there may be transient, small-scale structures that do not show up in the model. These structures can change the particle trajectory significantly. Therefore, the trajectories shown in the manuscript does not bear much useful information in explaining the multiple pressure peaks.

Line 537-538: '. . . indication of enhanced electric and magnetic shielding in the observations': How can you which of these two is effective from observation? As I commented above, the electric shielding may be not very effective for the energy range considered by the authors.

Figure 2a: Which MLT is this panel showing? Figure 2b: Which radial distance is this panel showing?

---

## Referee Comment (RC6) · Anonymous Referee #1 · 17 Aug 2018

My comment is simple: How did the authors calculate the plasma pressure?

The following is the procedure that I am currently understanding. First of all, please make sure if my understanding is correct.

1. For the TWINS results, the authors obtained the differential flux $F$ from ENA images. For CIMI, the authors calculated the differential flux $F$. $F$ has units of the number of ions/(unit energy · unit time · unit area · unit solid angle).

2. The authors calculated the pressure terms by integrating $F$ with respect to energy

and pitch angle.

$$P_\perp = \int d\cos\alpha \sin^2\alpha \int dE \sqrt{\frac{2E}{m}} F, \tag{1}$$

$$P_\parallel = 2 \int d\cos\alpha \cos^2\alpha \int dE \sqrt{\frac{2E}{m}} F. \tag{2}$$

Now, I realized that the confusion comes from the definition of $F$. Eqs. (1) and (2) will be understandable if $F$ is the velocity distribution function, NOT differential flux! The velocity distribution function, which is the number of particles in 6-dimensional space, is defined by

$$F \equiv \frac{dN}{d^3\mathbf{x} d^3\mathbf{v}},$$

where $N$ is the number of particles, and $v$ is velocity. The relationship between the velocity distribution function $F$ and the differential flux $j$ is given by

$$F = \frac{m^2}{2E} j.$$

Using this relationship, Eqs. (1) and (2) yield

$$P_\perp = \int d\cos\alpha \sin^2\alpha \int dE \sqrt{2Em} j, \tag{3}$$

$$P_\parallel = 2 \int d\cos\alpha \cos^2\alpha \int dE \sqrt{2Em} j. \tag{4}$$

Eqs. (3) and (4) are consistent with Eqs. (7) and (8) of De Michelis et al. (1997) who use the symbol $J$ to represent the differential flux. Hereinafter,

I would like to define the terms $F$ and $j$ to be the velocity distribution function and the differential flux, respectively, to avoid confusion. I would appreciate if the authors make sure which equations, (1)-(2), or (3)-(4), the authors used to calculate the pressure.

In the second reply, the authors stated that the plasma pressure was calculated by

$$P_\perp = \int p_{eq} d\cos\alpha \sin^2\alpha, \tag{5}$$

$$P_\| = 2\int p_{eq} d\cos\alpha \cos^2\alpha, \tag{6}$$

$$p_{eq} = \frac{2\pi}{m}\int EjdE. \tag{7}$$

Although Eqs. (5)-(7) are different from Eqs. (1)-(2) and Eqs. (3)-(4), the authors state that the change of the equations does not affect the results. Why? Does it mean that the authors did not use these equations to calculate the pressure? Does Eqs. (5)-(7) include typographical error? I may misunderstand something, but I would appreciate very much if the authors answer these questions.

---

## Author Comment (AC5) · 17 Aug 2018

My comment is simple: How did the authors calculate the plasma pressure?

The following is the procedure that I am currently understanding. First of all, please make sure if my understanding is correct.

1. For the TWINS results, the authors obtained the differential flux $F$ from ENA images. For CIMI, the authors calculated the differential flux $F$. $F$ has units of the number of ions/(unit energy · unit time · unit area · unit solid angle).

   Yes that is correct.

2. The authors calculated the pressure terms by integrating $F$ with respect to energy and pitch angle.

[Figure]

$$P_\perp = \int d\cos\alpha \sin^2\alpha \int dE \sqrt{\frac{2E}{m}} F, \qquad (1)$$

$$(2) \quad P_\| = 2 \int d\cos\alpha \cos^2\alpha \int dE \sqrt{\frac{2E}{m}} F.$$

We can only apologize to the Referee. There was a typing error in the equations we sent in our previous reply. The factor in the integral should be $\sqrt{2mE}$ . There also is a factor of $2\pi$ from the integral over the gyrotropic angle. The paragraph in the proposal is now

The pressure anisotropy shown in Figure 3 is defined as

$$A = \frac{P_\perp - P_\|}{P_\perp + P_\|}$$

with

$$\begin{Bmatrix} P_\perp \\ P_\parallel \end{Bmatrix} = 2\pi \int_{-1}^{+1} d\cos\alpha \begin{Bmatrix} \sin^2\alpha \\ 2\cos^2\alpha \end{Bmatrix} \left( \int_0^\infty dE \sqrt{2mE}\, F(E,n,\cos\alpha) \right)$$

where $\alpha$ is the ion pitch angle, $E$ is the ion energy, $n$ is the ion density, $m$ is the ion mass and $F(E,n,\cos\alpha)$ is the number flux per unit area, energy, time, steradian. This definition is derived from Braginskii (1965) and is consistent with previous formulations, e.g., Lui et al. (1987).

The units are now $steradians * E\sqrt{m^2 v^2}\, \dfrac{1}{El^2 t * steradians} = E\dfrac{mv}{mv^2 t} = \dfrac{E}{l^3}$ , i.e., energy/vol as it should be.

comment

Now, I realized that the confusion comes from the definition of $F$.

That is exactly correct.

[Figure]

Eqs. (1) and (2) will be understandable if $F$ is the velocity distribution function, NOT differential flux!

I am not sure what you mean by "differential flux". It is my understanding that one can have energy flux, number flux, charge flux, etc either per velocity, per energy, etc.

It is true that $f$ is often the used for the velocity distribution function. That is not what $F$ is the equation above and in the paper.

The velocity distribution function, which is the number of particles in 6-dimensional space, is defined by

$$F \equiv \frac{dN}{d^3\mathbf{x}\,d^3\mathbf{v},}$$

where $N$ is the number of particles, and $v$ is velocity.

Yes.

The relationship between the velocity distribution function $F$ and the differential flux $j$ is given by

$$F = \frac{m^2}{2E} j.$$

Using this relationship, Eqs. (1) and (2) yield

$$P_\perp = \int d\cos\alpha \sin^2\alpha \int dE \sqrt{2Em} j, \tag{3}$$

$$P_\parallel = 2 \int d\cos\alpha \cos^2\alpha \int dE \sqrt{2Em} j. \tag{4}$$

Eqs. (3) and (4) are consistent with Eqs. (7) and (8) of De Michelis et al. (1997) who use the symbol $J$ to represent the differential flux.

Yes, the corrected equations above are exactly as you say. If all that is needed to make it clear is to change $F$ to $j$, we have no problem with that.

Hereinafter, I would like to define the terms $F$ and $j$ to be the velocity distribution function and the

differential flux, respectively, to avoid confusion. I would appreciate if the authors make sure which equations, (1)-(2), or (3)-(4), the authors used to calculate the pressure.

[Figure]

We want the function in the integral to per unit energy. That is not what we would call a "velocity distribution".

In the second reply, the authors stated that the plasma pressure was calculated by

$$P_\perp = \int p_{eq} d\cos\alpha \sin^2\alpha, \tag{5}$$

$$P_\parallel = 2\int p_{eq} d\cos\alpha \cos^2\alpha, \tag{6}$$

$$p_{eq} = \frac{2\pi}{m}\int EjdE. \tag{7}$$

Although Eqs. (5)-(7) are different from Eqs. (1)-(2) and Eqs. (3)-(4), the authors state that the change of the equations does not affect the results. Why? Does it mean that the authors did not use these equations to calculate the pressure? Does Eqs. (5)-(7) include typographical error?

Honestly, we do not remember an equation of mine with a $j$ in it. We would not say that Eqs. (5-6) contain typographical errors. We would say they were ill-defined and unclear. We appreciate your efforts to make them clear. At this point, we think that they are at least well-

defined and describe appropriately the equations we used to calculate the anisotropy measurements and simulations we report in the paper.

I may misunderstand something, but I would appreciate very much if the authors answer these questions.

Given the unclear definitions we presented originally and the mistakes made in the equation we sent in our earlier reply, it is reasonable that you have not understood. To the best of our knowledge, the equations are now correct and well-defined.

To summarize, we want to use #ions per unit energy*area*time*steradians in the integral definition of the parallel and perpendicular pressure.

We have tried and will gladly continue to try to answer your questions until you are satisfied.

[Figure]

**ANGEOD**

---

## Referee Comment (RC7) · Anonymous Referee #1 · 18 Aug 2018

The authors answered my question properly. Everything is now clear. I have no additional comments or concerns. I recommend this paper for possible publication in Annales Geophysicae.

---

## Referee Comment (RC8) · Anonymous Referee #2 · 6 Sep 2018

The authors have addressed all my comments so I recommend this manuscript to be accepted.

---

## Author Comment (AC6) · 6 Sep 2018

This manuscript shows comparisons between models and observations of the pressure peaks in the inner magnetosphere during a storm event. The comparison reveals both consistency and significant differences between the observations and model predictions. The authors discussed the possible cause of the difference (i.e., the missing transient structures in the

[Figure]

Printer-friendly   version

simulation). The results of this manuscript are important for future improvement of models. However, there a few points that I would suggest the

authors to address before I recommend the manuscript for publication:

- Line 276: varies -> vary

Done. Thanks.

- Line 399-400: The authors start the sentence with both electric and magnetic shield-

ing but only explain magnetic shielding (gradient curvature drifts) in the later half of the sentence. The electric shielding is caused by the closure of region 2 current through the ionosphere, which creates a Peterson current, and thus electric field at lower latitudes than the region 2 current. This electric field, when mapped to the inner magnetosphere, cancels the original cross-tail electric field, so particles cannot ExB drift closer to Earth (see, e.g., Jaggi and Wolf, 1973). The electric shielding is more effective for low-energy particles. I do not think it is very important for the energy range which the authors are interested in.

Referee #1 made it clear to us that our use of the term magnetic shielding was not precise. The term has another meaning. So we have eliminated the term and replaced it with "spatially-localized, short-duration injections".

Again in response to comments by Referee #1, this paragraph has been significantly revised as follows: (We have added a reference to Jaggi and Wolf (1973) as suggested by Referee 2.)

*Injections from the plasma sheet are thought to be the primary source of ring current protons in the inner*

*magnetosphere, i.e., those that are observed by TWINS. Electric and magnetic fields determine the*

*ultimate path of the injected ions, i.e., whether they reach locations close enough to the Earth where the*

*magnetic gradient and curvature drifts are strong enough to exceed the electric drift forming the ring*

[Figure]

*current or whether they drift out to the magnetopause. The locations of the partial pressure peaks from the CIMI/RCM and the CIMI/Weimer 2K simulations and the TWINS observations during the 4-day period, 07-10 September 2015, show that the peaks are usually in the dusk/midnight sector. (See Figure 2b) This phenomenon is consistent with analysis of data at geosynchronous orbit (Birn et al., 1997). Nevertheless the TWINS observations show partial pressure peaks that are often at larger radii than the CIMI simulations, even when they are in the dusk/midnight sector (See Figure 2a.). The fact that the CIMI/Weimer peaks are generally closer to dusk than the CIMI/RCM. (See Figure 2b.) is consistent with simulations reported by Fok, et al. (2003). The TWINS MLT locations are closer to midnight and in the midnight /dawn sector more frequently than the CIMI results. This suggests that there are often enhanced electric shielding and effects from localized and short time injections that are not present in the CIMI simulations. To understand how the electric shielding works to affect the paths of the injected*

[Figure]

*particles, we note that the convection electric field from the solar wind is mapped into the magnetosphere along open field lines into the polar ionosphere. It is then shielded from penetrating to lower latitudes and therefore further into the inner magnetosphere by the Birkeland region 2 currents driven by pressure gradients in the ring current. See for example Jaggi and Wolf (1973). During geomagnetic storms when there is a sharp turn in the z-component of the interplanetary magnetic field (IMF) from negative to positive (See row 2 of Figure 1.), the accompanying electric field in the ionosphere associated with the Region 2 currents can produce what is referred to as over-shielding. There are also neutral disturbance dynamo electric fields in the ionosphere that affect electric shielding. Localized and short time injections may contribute to the complexity of these effects.*

As to the energy dependence of the effect of the electric field, it is true that for low energies where the magnetic drifts are small, the electric field is dominant. But it has been shown by Fok et al (2003) that a self-consistent electric field in place of the Weimer electric field model

moves the simulated peak of ions observed by IMAGE/HENA from the dusk side of midnight to the dawn side where it is observed. Thus it is clear that it does have an effect on the pressure in the energy we measure and simulate

- Line 455, and Line 547-548: 'parallel pitch angle anisotropy … first adiabatic invariant as they enter the inner magnetosphere': The conservation of first adiabatic invariant says that when a particle moves to a stronger magnetic field, it will have more perpendicular energy. Thus, the perpendicular anisotropy should increase instead of the parallel.

The Referee is correct. That was a mis-statement. That has been replaced by the following:

. *As they are accelerated while conserving the first adiabatic invariant to enter the region observed by TWINS, i.e. an outer radius of 8 R$_E$, their pitch angle distributions become parallel because the energy increase exceeds what can be absorbed in the perpendicular pitch angles while still conserving the first adiabatic invariant. One mechanism for reducing the parallel anisotropy is wave-particle interactions which are not included in the CIMI simulations..*

The key point is that the particles are increasing their energy as they enter from the tail. This is illustrated in Figure 10.

[Figure]

- Line 512-Line 527: This paragraph makes a strange comparison. To find the origin of the multiple pressure peaks, the authors uses particle tracing in the model, which does not have the multiple pressure peaks. As the authors said, the reason why the model cannot reproduce the observed multiple peaks is that there may be transient, small-scale structures that do not show up in the model. These structures can change the particle trajectory significantly. Therefore, the trajectories shown in the manuscript does not bear much useful information in explaining the multiple pressure peaks.

The Referee is correct in saying that the model fields that we use for the particle tracing is not one that necessarily produced the multiple peaks. The idea is that it might have if the input across the outer boundary at 10 RE in CIMI simulations had included non-isotropic, spatially localized and short-time dependent injections.

Line 537-538: '… indication of enhanced electric and magnetic shielding in the observations': How can you which of these two is effective from observation? As I commented above, the electric shielding may be not very effective for the energy range considered by the authors.

As stated above we think it is clearer to speak of "enhanced electric shielding and/or spatially-localized, short-duration injections". The Referee is correct that the relative importance of the two effects cannot be determined from observations of the type we show here. That is why we are trying to compare observations with simulations.

[Figure]

As for the energy dependence of the electric shielding, the fact that it is important for more than just low energies has been demonstrated by Fok et al (2003).

[Figure]

Figure 2a: Which MLT is this panel showing? Figure 2b: Which radial distance is this panel showing?

It is showing the location, radial distance and MLT, of the main peak. The one marked by the star in the figures. We will add a statement to that effect.

---

## Author Comment (AC7) · 21 Sep 2018

I am new to this process, so I am unsure how to proceed.

We have responded positively to all the referees' comments. Our responses are correlated with each referee's posting. Each referee has responded saying they are satisfied. If there is something else I am supposed to do, please let me know.

J. D. Perez

---

## Author Comment (AC8) · 28 Sep 2018

This is my second attempt at making this response.

The last communication from Referee #1 states that the authors have answered all of his questions.

We wanted to submit a summary of the discussion between the authors and the referee #1 noting the location of the changes in document developed in response to comments from both referees.

It does not seem to be possible to upload 2 documents, so they are now combined into

one for referee #1.

We sincerely hope this is what is needed.

Please also note the supplement to this comment:
https://www.ann-geophys-discuss.net/angeo-2018-64/angeo-2018-64-AC8-supplement.pdf
* * *
[Figure]

**Supplement:**

[GENERAL COMMENTS]

This paper presents the equatorial plasma pressure distributions obtained by the TWINS observation and by the drift kinetic simulation CIMI for the moderate storms of 7-10 September 2015. The general features of the plasma pressure in the inner mag- netosphere are similar to each other, whereas some differences are found in terms of peak location, anisotropy, and spatial distribution. The authors attributed the dif- ferences to the shielding effect and spatially-localized, short-duration injections of hot plasma.

The direct comparison between a sophisticated observation and an advanced drift kinetic equation is highly valuable, and is promising to overcome the difficulties arising from in-situ satellite observations. The provided data is basically very interesting, and I admire the authors' efforts to derive the pressure and anisotropy.

**The authors thank the referee for the thoughtful and helpful comments. We respond positively below to each question and comment individually. Nevertheless, we cannot answer every question posed by referee with a full, unambiguous explanation. Such requires extensive investigations that are underway but are beyond the scope of this paper. We believe what many have said, i.e., good research raises at least as many questions as it answers.**
   **Our responses are shown in bold font for ease of distinguishing our responses and the referee's comments.**

However, I have 3 major concerns as follows:

First, the physical interpretations made by the authors are unclear. Because of the lack of proper interpretations, I cannot catch new scientific knowledge or insights in the current version of the manuscript.

**We certainly hope that our specific responses allow the referee and other readers to "catch new scientific knowledge or insights. . . "**

Secondly, the reliability of the plasma pressure obtained by TWINS is also unclear. The spectral shape of the ion flux is almost the same at 4 different points, which seems unlikely to occur.

 **The intent of showing the spectral shapes in Figure 10 was not to present an extended discussion of the spectra but rather to indicate that were two peaks, one at low energy, i.e., below 20 keV, and a second near 40 keV and to illustrate why the paths of 40 keV particles are shown. We are not sure that this is unlikely to occur. Details of the magnitude and shape of the energy spectra will be addressed in an extensive study that is underway.**

 **As to the overall reliability of the plasma pressure, that is a difficult question to answer definitively and quantitatively. In the description of the methodology used to obtain the ion pressure from the TWINS ENA images, we have referenced the extensive testing of the ion distributions obtained from**

the TWINS ENA images. (See Section 2.2, lines 190 ff in the version of the paper where corrections are accepted and lines 194 ff in the version of the paper where corrections are marked.)

Thirdly, the plasma pressure mentioned in this paper is "partial" so that the "true" distribution of the plasma pressure would be different. Careful description is needed when the authors intend to say the distribution of the pressure.

**The referee is correct. What we show in this paper are "partial" pressures. The intent is to have the TWINS and CIMI results consider as nearly as possible a similar energy range. In the revised paper, all references to the particular pressures from TWINS and CIMI are now designated as partial pressure.**

[SPECIFIC COMMENTS]

1.On the interpretations. The authors concluded that the difference between the ob- servation and the simulation can be best explained by enhanced electric and magnetic shielding and/or spatially-localized, short-duration injections. First of all, please explain the meaning of the electric and magnetic shielding in more detail. Most of the readers may not understand the meaning of it. The electric shielding is supposed to result from the ionospheric electric field redistributed by the Region 2 field-aligned current.

**Excellent suggestion. We include in the revised version an explanation as to what we mean by electric shielding. [See Lines 436-445]**

What is the magnetic shielding?

**"magnetic shielding" was an improper term. Better to say enhanced electric field shielding and/or induction electric fields caused by spatially-localized, short duration injections. The term is omitted in the revised version.**

What is the expected effect of the shielding on the pressure distribution and pressure anisotropy?

**It has been demonstrated that the electric shielding can affect the ring current morphology. [See Lines 90-94 in the revised document.] The purpose of this paper is to show much more explicitly what are the expected effects. We are currently undertaking a project to couple the CIMI code in the inner magnetosphere with a 3D hybrid code that simulates the rest of the magnetosphere. We expect the results to address these issues in even greater detail.**

CIMI/RCM takes into account the shielding. What physical processes or parameters does CIMI/RCM need to consider properly to explain the observations?

**The CIMI simulations presented in this paper do not have localized injections into the inner magnetosphere. This has been demonstrated to have an effect on the pressure distributions. [See Lines 524-526 in the revised document.] While the CIMI simulations in this paper do include some contributions from induction electric fields, the Tsyganenko magnetic field is not updated on short enough time scale to capture all of the induction electric fields. We, of course, do not know whether the steps we are taking to provide localized and short term injection effects will answer all the questions.**

Have the authors tested CIMI/RCM with different conditions/parameters to explain the observations?

**There have been investigations that demonstrate that changing the input at the boundary of the CIMI simulations does impact the ring current morphology. Also it has been shown that injecting non-isotropic pitch angle distributions impacts the parameter of the ring current. [See Lines 507-509 in the revised document] The authors have not tried, however, to reverse engineer the input to the CIMI simulations presented in this paper to attempt to match the data.**

(1) On reliability of the plasma pressure. In Figure 10, the differential fluxes of the ions are shown as a function of energy at 4 points. The intensity of the flux is different but the spectral shape is almost the same with each other.  Why is the spectral shape of the flux almost the same at the 4 points?  According to in-situ observations, the spectral shape of the flux depends on L-value and magnetic local time (e.g., Milillo et al., 2001, 10.1029/2000JA900158), so that it seems kquite unlikely to be the same spectral shape at 4 points. Please explain the validity of the spectral shape of the flux and the plasma pressure distribution presented in this paper.

**We reference the published examples of validation of all the features of the ion distributions through comparisons with in-situ measurements. [See Lines 191-202.] These examples were chosen because the satellites happen to be in the right place at the right time to see the features of interest.  It is true, of course, that such comparisons do not guarantee the complete validity of the current results.**
**The main reason for showing the measured energy spectra was to show why the paths of 46 keV ions were used to display the location and time of the injections of the ions that had 46 keV energy at the peaks.  What the authors think has the relevant validity in this case is the high energy a low energy peaks in the spectra.**
**The outstanding work by Millilo et al , 2001 will make an important contribution to our current investigation into the details of the energy spectra during geomagnetic storms during the TWINS mission.  The fact that the average results presented in their paper is for AE < 100 nT does not directly invalidate the results presented here.**
**Finally, as was stated above, the focus of this paper is not the details of the energy spectra.  If the referee would prefer, we could remove them from Figure 10.**

Secondly, please explain the expected effect of spatially-localized, short-duration injections on the pressure and anisotropy.

**As described above and is illustrated to some extent in previously referenced work, the authors expect that spatially-localized, short-duration injections will impact the spatial and temporal locations of the pressure peaks.  We also expect it to be a key factor in explaining the observation of multiple peaks in the ring current.**

Have the authors modeled spatially-localized, short-duration injections to explain the observations?

**As stated above, it has been demonstrated that spatially localized injections can affect the ring current morphology, but we have not tried to match the observations for this particular storm without some experimental or theoretical guidance. There is an ongoing investigation to couple CIMI with a 3D hybrid simulation of the injections from the tail explicitly intended to address this question.**

Thirdly, please explain the reason why the CIMI result always shows parallel anisotropy of the plasma pressure in the dawn-midnight-dusk region. The pressure anisotropy is largely different from the observations. Detailed explanation is needed.

**An explanation is given in the revised paper. [See Lines 490-496 in the revised document.] Whether this is the complete explanation is uncertain at this time. The authors expect to develop a more definitive explanation as part of an ongoing investigation to couple CIMI with a 3D hybrid simulation of the injections from the tail.**

(2) On the plasma pressure. I suppose that the plasma pressure was calculated from integration of the differential flux over the energy range from 2.5 keV to 97.5 keV. The energy range is probably insufficient to cover all the ions trapped in the inner mag- netosphere because the ions with energy greater than 100 keV is also known to con- tribute to the plasma pressure (energy density) largely (e.g., Smith and Hoffman, 1973, 10.1029/JA078i022p04731; Williams, 1983, 10.1016/0032-0633(81)90124-0). If the high energy ions remained during these storms, there would be another peak of the pressure, which may stay at L  2.5 – 3.0. I recommend discussing possible impacts of the high energy ions (>100 keV) on the conclusion. I also recommend emphasiz- ing that the plasma pressure distribution is "partial" so that the pressure distribution is incomplete.

**The referee is correct, the plasma pressure presented in this paper should be referred to as partial pressure because it was calculated by integrating from 2.5 to 97.5 keV. The paper by Smith and Hoffman, 1973 certainly shows that higher energies can make significant contributions to the energy density (pressure).  It is to be noted, however, that they say**

**"To contrast the development of the ring current for the two storms, we now consider those protons (1- to 138-kev protons were used) which contribute substantially to the storm-time ring current. While protons in this energy regime contribute only 20% or less to the total energy density out to L -• 4 during magnetically quiet periods, their enhancement during magnetic storms, combined with a depletion of protons with energies greater than about 170 kev, make them the dominant (greater than 90%) contributors to the storm-time energy densities."**

**The paper is referenced. [See Line 179 in revised document.)**

**The paper by William, 1983, describes the state of observations at that time with the conclusion, "It is found that the ring current energy density composition still has not been observed."**

**The authors wholeheartedly agree with the recommendations of the referee.**

[MINOR COMMENTS]

Introduction: I recommend citing papers related to plasma pressure distribution and anisotropy observed by satellites, for example, De Michelis et al. (1999, 10.1029/1999JA900310), Ebihara et al. (2002, 10.1029/2002GL015430), and Lui (2003, 10.1029/2003GL017596).

**Definitely.  The authors apologize for not recognizing these papers. [See Lines 65, 70, 72  in revised document.]**

Line 47-57: Simulation results with different electric field and/or magnetic field models have been conducted by Angelopoulos et al. (2002, 10.1029/2001JA900174) and Ebihara et al. (2004, 10.5194/angeo-22-1297-2004).

**Most definitely.  The authors thank the referee for pointing us to these papers. [See Lines  55, 61 in revised document.]**

Line58-63: This paragraph seems not to provide information. What key spatial features do Wang et al. (2011) find?

**The authors think it is somewhat harsh to say that the paragraph [Lines 65- does not provide any information. In particular reference to the Wang et al. (2011) paper, the paper presents extensive data and comparisons with RCM modeling, but they are based upon statistical averages of events. The result from Wang et al (2011) that is relevant to this paper is what is stated in the last sentence of this paragraph. It is based upon the last sentence of the Abstract and the first sentence of the Summary.**

Line 311: The equatorial pressure p_eq is difficult to understand. Please explain how to derive p_eq.

**The authors are not sure why p_eq is difficult to understand. It is pressure at the equator as a function of position and pitch angle. It is the standard definition of pressure, i.e., the energy density of the ions, the integral of the distribution function times the velocity squared.**

**What follows is an exchange between the referee and the authors that clarified the original equations. The referee's comments are shown in different colors reflecting the order in which they entered the discussion. The authors responses are in bold black. The paragraph now reads as shown in Lines 333-340 of the revised document.**

The authors answered my comments properly except for one thing on the equatorial pressure (Line 311 in the first version of the manuscript). The authors stated that peq is pressure at the equator as a function of position and pitch angle, and that it is the standard definition of pressure. If it is the standard definition of pressure, please cite relevant reference. The reason why I am asking is that readers may be eager to know how the authors obtained the terms,P⊥ and P||. Here, I assume that P⊥ and P||are the pressure tensor components in the perpendicular and parallel components, respectively. Lui et al. (1987, 10.1029/JA092iA07p07459) show the equations to calculate P⊥ and P|| as a function of velocity v and the velocity distribution function f (Eqs. 2 and 3 in Lui et al., 1987). The velocity distribution function f is associated with the differential flux that is directl ymeasurable. What is the relationship between peq and the measured value (probably differential flux derived from the ENA observation)?

**The Referee is correct, the equation as presented is unclear. Somehow the full response to the original comment was lost.**

**The pressure anisotropy shown in Figure 3 is defined as**

$$A = \frac{P_\perp - P_\parallel}{P_\perp + P_\parallel}$$

**with**

$$P_\perp = \int_{-1}^{+1} p_{eq}(\alpha) \sin^2 \alpha \, d\cos\alpha \quad \& \quad P_\parallel = 2\int_{-1}^{+1} p_{eq}(\alpha) \cos^2 \alpha \, d\cos\alpha$$

where α is the pitch angle and $p_{eq}$ is the equatorial pressure as a function of location and pitch angle which was obtained from the energy dependent number flux deconvolved from the TWINS ENA images, i.e.,

$$p_{eq} = \frac{2\pi}{m} \int_0^\infty E \; f(E, n, \cos\alpha) \; dE$$

where $f(E, n, \cos\alpha)$ is the number of ions per unit area, energy, and steradian.  This definition is derived from Braginskii (1965) and is consistent with previous formulations, e.g., Lui et al. (1987).

The authors now present the meaning of the equatorial pressure peq as peq = 2π mREfde, where m is mass, E is energy, and f is the number of ions per unit area, energy and steradian. First of all, I am unsure if f corresponds to the so-called differential number flux that is the number of ions per unit area, energy, time and steradian. If so, I have further comment. The definition of the perpendicular and parallel pressure (P⊥ and P||) is as follows. P⊥ = 1 2Rmv2 sin2 αFdv and P|| = Rmv2 cos2 αFdv, where F is the velocity distribution function. These equations can be derived from the original definition of pressure (probably Braginskii (1965) provided, too). Lui et al. (1987) also present these equations. The velocity distribution function is given by F = m v2 f. Substituting this into above equations, I have P⊥ = RRπv2mf√E sin2 αd(cosα)dE = RRπv2mf√EdE sin2 αd(cosα) = Rpeq sin2 αd(cosα)and P|| =RR2πv2mf√E cos2 αd(cosα)dE =R2peq cos2 αd(cosα), where peq =Rπv2mf√EdE. It seems that the definition of peq is different from the authors'. The same equation is found in Eqs (7) and (8) in De Michelis et al. (1997, doi:10.1029/96JA03743). Maybe I misunderstand, but I would like to make it clear. I suggest avoiding the term 'equatorial pressure' because this term is confusing and misleading. The above equations can be applied for everywhere, not restricted in the equatorial plane.

The question seems to revolve around 2 areas of confusion.

First, the use of the subscript "eq" to indicate that it is the equatorial pressure.  The purpose was to communicate that it is at the equator that we actually calculate the pressure anisotropy.  We agree, however, that this may be confusing so we agree to leave it out.

Second the definition of the symbols *f* and *F* in the equations.  We suggest that to try to avoid this confusion, we suggest the following:

The pressure anisotropy shown in Figure 3 is defined as

$$A = \frac{P_\perp - P_\parallel}{P_\perp + P_\parallel}$$

**with**

$$\begin{Bmatrix} P_\perp \\ P_\parallel \end{Bmatrix} = \int_{-1}^{+1} d\cos\alpha \begin{Bmatrix} \sin^2\alpha \\ 2\cos^2\alpha \end{Bmatrix} \left( \int_0^\infty dE \sqrt{\frac{2E}{m}} F(E, n, \cos\alpha) \right)$$

**where α is the ion pitch angle, *E* is the ion energy, *n* is the ion density, *m* is the ion mass and *F(E,n,cos***

**α)* is the number flux per unit area, energy, time, steradian. This definition is derived from Braginskii**

**(1965) and is consistent with previous formulations, e.g., Lui et al. (1987).**

The authors revised the equation for the pressure terms. However, the equation seems to be different from the equation given by De Michelis et al. (1997, http://doi.wiley.com/10.1029/96JA03743) by a factor of πm. I am curious to know the reason why the equation is different. The definition of the plasma pressure is changed. Does this change have any impact on the result? I suppose that lower energy protons may have more impact on the pressure. I recommend removing n from F(E,n,cosα) because F is an arbitrary function and n is an independent variable.

**We very much appreciate the comments that have been made by the Referee.  They have certainly made our paper better. But at this time, it seems that we having trouble communicating.  We are not sure how to respond.  Please see comments below: (The Referee's comments are repeated in italics.)**

The authors revised the equation for the pressure terms. However, the equation seems to be different from the equation given by De Michelis et al. (1997, http://doi.wiley.com/10.1029/96JA03743) by a factor of πm. I am curious to know the reason why the equation is different.

**I assume that the referee is referring to Eqs. (7) and Eqs (8) in De Michelis et al. (1997).  Let's look at the unit of those eqs. For the pressure to have the correct units of energy per unit volume, the units of the "differential flux intensity", *J* in Eqs. (7) and (8) must be 1/(vol * m * v).  The reason for the different factors is that *J* is not the flux we have in our equation.  Our flux, *F(E,n,cosα),* as it says in the text, has units of #ions/(energy\*time\*area\*steradian).  One can check the units of the equation we have in the paper and they come out to be pressure.  In fact if one substitutes the proper equation for a Maxwellian distribution into the equation in the paper, i.e.,**

$$F(E, n, \cos\alpha) = \frac{n}{\sqrt{2m}(\pi T)^{3/2}} E\, e^{-E/T}$$

**and performs the integrals, the result is *nT* , precisely what one expects for a Maxwellian.**

The definition of the plasma pressure is changed*.*

**The definition of the plasma pressure has not changed. We have been discussing a general definition of pressure. That has not changed, just the way it is presented has been made clearer with the help of the Referee.**

Does this change have any impact on the result? I suppose that lower energy protons may have more impact on the pressure*.*

**We assume this is in reference to the previous comment regarding the change in the definition of the pressure. We assume that the referee is referring to the fact that the pressure we calculate and present as results is the partial pressure, i.e., it is integrated from 2.5 to 97.5 keV for TWINS and 1 to 133 keV for CIMI. The referee correctly requested that we distinguish the pressure calculated in this paper as the partial pressure and we have done so. There is no change that would impact the results in this paper.**

I recommend removing n from F(E,n,cosα) because F is an arbitrary function and n is an independent variable.

**This comment may somehow be at the heart of the miscommunication that we are having at this time. Yes, it is in some sense arbitrary, i.e., in the expression for the pressure, it is whatever it is in a particular physical situation. In what we are presenting, however, the *F(E,n,cosα)* is definitely not an arbitrary function. For the TWINS results, it is what is obtained from the ENA images. For CIMI, it is what is obtained from the simulations. As stated above it has units of #ions/(energy\*time\*area\*steradian). We feel that it makes the most sense to express the pressure in terms of what it is obtained, i.e., from the measurements and simulations. It is then integrated as expressed in the formulas to obtain the pressures we present. We might also note that previous publications of TWINS and CIMI results have shown the average of *F(E,n,cosα)* over pitch angles as a function of energy.**

My comment is simple: How did the authors calculate the plasma pressure? The following is the procedure that I am currently understanding. First of all, please make sure if my understanding is correct.

1. For the TWINS results, the authors obtained the differential flux F from ENA images. For CIMI, the authors calculated the differential flux F. F has units of the number of ions/(unit energy·unit time·unit area·unit solid angle).

**Yes that is correct.**

2. The authors calculated the pressure terms by integrating F with respect to energy and pitch angle.

$P\perp = Z\, dcos\alpha sin2\ \alpha Z\ dEr2E\ m$

F, (1)

$P|| = 2Z\, dcos\alpha cos2\ \alpha Z\ dEr2E\ m$

**We can only apologize to the Referee.  There was a typing error in the equations we sent in our previous reply.  The factor in the integral should be $\sqrt{2mE}$ .  There also is a factor of 2π from the integral over the gyrotropic angle. The paragraph in the proposal is now**

   **The pressure anisotropy shown in Figure 3 is defined as**

$$A = \frac{P_\perp - P_\parallel}{P_\perp + P_\parallel}$$

**With**

$$\left\{ \begin{array}{c} P_\perp \\ P_\parallel \end{array} \right\} = 2\pi \int_{-1}^{+1} d\cos\alpha \left\{ \begin{array}{c} \sin^2\alpha \\ 2\cos^2\alpha \end{array} \right\} \left( \int_0^\infty dE\, \sqrt{2mE}\ F(E,n,\cos\alpha) \right)$$

**where α is the ion pitch angle, E is the ion energy, n is the ion density, m is the ion mass and F(E,n,cos α) is the number flux per unit area, energy, time, steradian. This definition is derived from Braginskii (1965) and is consistent with previous formulations, e.g., Lui et al. (1987).**

**The units are now** $steradians * E\sqrt{m^2 v^2}\ \dfrac{1}{El^2 t * steradians} = E\,\dfrac{mv}{mv^2 t} = \dfrac{E}{l^3}$ **, i.e., energy/vol as it should be.**

Now, I realized that the confusion comes from the definition of F.

**That is exactly correct.**

Eqs. (1) and (2) will be understandable if F is the velocity distribution function, NOT differential flux!

**I am not sure what you mean by "differential flux". It is my understanding that one can have energy flux, number flux, charge flux, etc either per velocity, per energy, etc.**

**It is true that $f$ is often the used for the velocity distribution function. That is not what $F$ is the equation above and in the paper.**

The velocity distribution function, which is the number of particles in 6-dimensional space, is defined by

$F = m2 \ 2Ej$.

Using this relationship, Eqs. (1) and (2) yield $P\perp =Z \ d\cos\alpha\sin^2 \alpha Z \ dEV2Emj$, (3) $P|| = 2Z \ d\cos\alpha\cos^2 \alpha Z \ dEV2Emj$. (4) Eqs. (3) and (4) are consistent with Eqs. (7) and (8) of De Michelis et al. (1997) who use the symbol J to represent the differential flux.

**Yes, the corrected equations above are exactly as you say. If all that is needed to make it clear is to change $F$ to $j$, we have no problem with that**.

Hereinafter, I would like to define the terms F and j to be the velocity distribution function and the differential flux, respectively, to avoid confusion. I would appreciate if the authors make sure which equations, (1)-(2), or (3)-(4), the authors used to calculate the pressure.

**We want the function in the integral to per unit energy. That is not what we would call a "velocity distribution".**

In the second reply, the authors stated that the plasma pressure was calculated by $P\perp =Z \ peqd\cos\alpha\sin^2 \alpha$, (5) $P|| = 2Z \ peqd\cos\alpha\cos^2 \alpha$, (6) $peq = 2\pi \ mZ \ EjdE$. (7) Although Eqs. (5)-(7) are different from Eqs. (1)-(2) and Eqs. (3)-(4), the authors state that the change of the equations does not affect the results. Why? Does it mean that the authors did not use these equations to calculate the pressure? Does Eqs. (5)-(7) include typographical error?

**Honestly, we do not remember an equation of mine with a $j$ in it. We would not say that Eqs. (5-6) contain typographical errors. We would say they were ill-defined and unclear. We appreciate your efforts to make them clear. At this point, we think that they are at least well- defined and describe appropriately the equations we used to calculate the anisotropy measurements and simulations we report in the paper.**

I may misunderstand something, but I would appreciate very much if the authors answer these questions.

**Given the unclear definitions we presented originally and the mistakes made in the equation we sent in our earlier reply, it is reasonable that you have not understood. To the best of our knowledge, the equations are now correct and well-defined.**

**To summarize, we want to use #ions per unit energy\*area\*time\*steradians in the integral definition of the parallel and perpendicular pressure.**

**We have tried and will gladly continue to try to answer your questions until you are satisfied.**

Line 489-496: Ebihara et al. (2009) also showed multiple peaks of the plasma pressure distribution in the inner magnetosphere by introducing temporal changes in the distribution function at the outer boundary of CRCM. It would be worth mentioning that the rapid changes in the distribution function in the plasma sheet could result in the multiple peaks of the plasma pressure**.**

**A sentence has been added pointing out that the model calculations did show multiple pressure peaks inside of 4 $R_E$.  [See Lines 534-535 in revised document.]**

Line 499 "But they do not provide incontrovertible evidence for the effects of spatially and temporally dependent injections into the inner magnetosphere." This sentence is difficult to understand.

**The sentence has been removed.**

Figure 10, caption: Please indicate the unit of the color bar (probably in keV), and pitch angle of the particle. What is the meaning of "Minimum – Maximum Energy for Each Path"?

**An explanation has been added. [See Figure 10 caption, Line 1035 of revised document.]**

Line 520, "Peak 5" Does it mean "Peak 4"?

**Yes.  It has been corrected. [See Line 563 of revised document.]**

Line 497-527: The spectral shape of the differential flux of the ions is almost the same at the 4 points. Please explain the validity of the differential flux derived from TWINS? At Peak 3, the ion is inaccessible from the outer boundary. I recommend tracing the ion trajectory backward in time by starting at slightly different points.

**I think we have addressed this issue in response to previous comments.  The main reason for presenting these spectra is to motivate showing the paths of the 46 keV ions.  As stated above, if the referee prefers, they can be removed.**

Line 546-548: "This is not unexpected as the ions are being injected into regions of higher magnetic field, and conservation of the first adiabatic invariant would predict the enhancement of parallel pitch angles." I cannot understand this meaning. Please explain the reason why the conservation of the first adiabatic invariant results in the pressure anisotropy dominated by the parallel component?

**A full explanation has been added. [See Lines 589-598 in revised document.]**

Line 548-550: "Nevertheless the parallel anisotropy is seen in the observations only during the main phase of the first storm. This is also an indication of stronger electric and magnetic shielding." Please explain the reason why the stronger shielding results in the parallel anisotropy?

**The statement has been changed to be consistent with responses to previous comments by the referee. [See Lines 589-598 in revised document.]**

The authors answered my question properly. Everything is now clear. I have no additional comments or concerns. I recommend this paper for possible publication in Annales Geophysicae

Dynamics of a Geomagnetic Storm on 7-10 September 2015 as Observed by TWINS and
                          Simulated by CIMI

Perez[1], Joseph D., James Edmond[1], Shannon Hill[2], Hanyun Xu[1], Natalia Buzulukova[3], Mei-
Ching Fok[3], Jerry Goldstein[4,5], David J. McComas[6] and Phil Valek[4,5]
[1]Auburn University, Auburn, AL 36849, USA
[2]Emory University, Atlanta, GA 30322, USA
[3]NASA Goddard Space Flight Center, Greenbelt, MD 20771, USA
[4]Southwest Research Institute, San Antonio, TX 78228, USA
[5]University of Texas at San Antonio, San Antonio, TX 78249, USA
[6]Department of Astrophysical Sciences, Princeton University, NJ 08540, USA
*Correspondence to*: J. D. Perez, perez@physics.auburn.edu

[revised manuscript text omitted]

nT.  Ebihara et al. (2002) compared statistically averaged data from POLAR/MICS (Wilken, et al., 1992) with simulations of proton drift paths using the Volland-Stern electric potential and found reasonable agreement.  Lui, et al. (2003) used the AMPTE/CCE-CHEM and MEPA

(McEntire et al., 1985) to construct the plasma pressure distribution over an extended energy range from 1 keV to 4 MeV. They found that the statistical pressure distribution obtained from the in-situ measurements differed from the results obtained from ENA images obtained from

IMAGE/HENA (Brandt et al., 2004).  Wang et al (2011) compared average spatial profiles of the

Time History of Events and Macroscale Interaction during Substorms (THEMIS) (Angelopoulos,

2008) in situ-observations with simulations using the Rice Convection Model (RCM) self- consistent electric and magnetic fields (Toffoletto et al, 2003). The agreement with key spatial features of the particle fluxes confirms the importance of the magnetic and electric transport in determining features of the ring current. With the advent of missions dedicated to energetic neutral atom (ENA) imaging, e.g., (1) the 3 instruments, LENA (T. E. Moore et al, 2000),

MENA (Pollock et al, 2000), and HENA (Mitchell et al, 2000) on board IMAGE (Burch, 2000), (2) the Energetic Neutral Atom Detector Unit (NUADU) (McKenna-Lawlor et al, 2005), and (3)

Two Wide-angle Imaging Neutral-atom Spectrometers (TWINS) (McComas et al, 2009a;

Goldstein and McComas, 2013; Goldstein and McComas, 2018), it became possible to test simulations against full images of the inner magnetosphere.

Fok et al (2003) compared simulations using the CRCM (Fok et al, 2001) model with ENA

images from IMAGE/MENA & HENA.  They were able to match the magnitude and trends of the observed Dst but not all of the short time variations. The empirical Weimer96  electric field model was not able to explain the fact that the peaks of the proton flux in the inner magnetosphere were in the midnight/dawn sector rather than the expected dusk/midnight sector during a strong storm on 12 August 2000, but the self-consistent CRCM electric field model did explain this feature.  They also used the MHD fields computed by the BATS-R-US (Block-

Adaptive-Tree Solar-wind Roe Upwind Scheme) (Groth et al, 2000) model to provide electric and magnetic fields and ion temperature and density at the model boundary (10 $R_E$) at the equator to model a large storm that occurred on 15 July 2000.  The simulated ENA images matched the general features of the HENA ENA images.

Buzulukova et al. (2010) studied the effects of electric shielding on ring current morphology by comparing the results of CRCM simulations from a moderate and a strong storm with ENA

images from TWINS and IMAGE/HENA.  The Tsy96 empirical magnetic field, the Weimer-

2000 electric potential model (Weimer, 2001) and the empirical Tsyganenko amd Mukai (2003)

model of the plasma sheet density and temperature were employed.  They achieved agreement between the magnitude and trends of the observed SYM/H and the simulated values for both storms, and were able to explain the post-midnight enhancements of the pressure due to electric shielding.  They did not include the effects of inductive electric fields or time dependence due to substorms.

Fok et al (2010) used ENA images from both TWINS1 and TWINS2 along with in-situ

THEMIS observations during a storm on 22 July 2009 to validate the CRCM simulations.  They found that, when a time-dependent magnetic field is included, the electric potential pattern is less twisted and the ion flux peak did not move as far eastward giving better agreement with the ENA

observations.

It is clear that present-day simulations are able to explain the general features of the observations of the ring current in the inner magnetosphere, both from in-situ measurements and in ENA images.  It is also clear that questions remain as to the contributions of various shielding mechanisms.  Self-consistent dynamic electric potentials give better results.  Inclusion of magnetic induction effects is also necessary for the best results.  But to date effects on short time scales, e.g., injections from sub-storms, bubbles, and bursty bulk flows have not been included in a self-consistent manner.

It is also important to note that the cases treated have been either statistical averages or single events in which there was no evidence for multiple peaks in the ring current pressure distribution.  The existence of multiple peaks, however, has been observed in data from the

AMPTE Charged Particle Explorer mission (Liu et al, 1987; Ebihara et al, 1985) and in ion distributions extracted from TWINS ENA images (Perez et al., 2015).

The science question to be addressed by this study is: Are there features in the global ring current pressure that are caused by enhanced electric shielding and/or spatially-localized, short- duration injections?   We present for the first time a direct comparison between simulations of ring current equatorial partial pressure and anisotropy distributions with the unique global images extracted from the TWINS ENA images.  We present cases in which the general characteristics of the observed partial pressure distribution are reproduced by the simulations and others in which the observed ion partial pressure peaks are at larger radius, in different MLT

sectors, and display multiple peaks that are not found in the simulations. We also compare for the first time global images of the pressure anisotropy extracted from the TWINS ENA images with the results of simulations using the Comprehensive Inner Magnetosphere Ionosphere (CIMI)

model (Fok et al., 2014).

In Sect. 2, we describe the measurement of the TWINS ENA images and the process by which ion partial pressures and anisotropy are extracted, and briefly discuss how this technique has been validated against in-situ measurements.  In Sect. 3, we describe the important aspects of the CIMI model, and how it has been compared with geomagnetic activity indices, in-situ measurements, and ENA images.  The particular storms on 7-10 September 2015, which are the focus of this study, are described in Sect. 4.  The comparison of results of the measurements and simulations are presented in Sect. 5.  They are discussed in Sect. 6.  Sect. 7 summarizes the results and the conclusions.

**2 Measurements**

**2.1 TWINS ENA Images**

The NASA TWINS mission of opportunity (McComas et al., 2009a; Goldstein and McComas,

2013, Goldstein and McComas, 2018) obtains ENA images of the inner region of the Earth's magnetosphere. The instrument concept is described in McComas et al. (1998).  Every 72 s with an integration (sweep) time of 60 s, full images are obtained. In this study, in order to obtain sufficient counts for the deconvolution process described in Sect. 2.2, the images are integrated over 15-16 sweeps.  This means data is collected for  ~15 min over an ~ 20 min time period. The energies of the neutral atoms span a range from 1-100 keV/amu.  In the images used in this study, the energy bands are such that $\Delta E/E =1.0$ for H atoms. In order to enhance the processed image, a statistical smoothing technique and background suppression algorithms described in detail in Appendix A of McComas et al. (2012) are employed.  This combined approach is an adapted version of the statistical smoothing technique used successfully for IBEX (McComas et al., 2009b) data.

**2.2 Ion Pressures**

For the comparison with simulation results using the CIMI program (See Sect. 3.), the spatial and temporal evolution of equatorial ion partial pressure and pressure anisotropy are routinely obtained from the TWINS ENA images. To extract this information from the ENA images, the ion equatorial pitch angle distribution is expanded in terms of tri-cubic splines (deBoor, 1978).

To fit the data and to obtain a smooth solution, the sum of normalized chi-squared and a penalty function derived by Wahba (1990) is minimized.  The penalty function is what produces the smoothness of the result (in the sense of a minimum second derivative), and the normalized chi- square is what ensures that the calculated image corresponds to the measured ENA image. This means that the spatial structure obtained in the equatorial ion partial pressure distributions is no more than is required by the observations (Perez et al, 2004).  In order to obtain pressures from the energy dependent ENA images, which are integrated over energy bands with widths equal to the central energy, e.g., 40 keV images are integrated from 20-60 keV, a technique using singular valued decomposition as described in Perez, et al., (2012, Appendix B) is employed.  The energy range included in the partial pressures presented in this paper is 2.5-97.5 keV, i.e., the energy range observed by TWINS.  It is to be noted that higher energies do make significant contributions to the total ring current pressure. (Smith and Hoffman, 1973)

In order to obtain the ion distributions from the ENA images, models for both the magnetic field and the exospheric neutral hydrogen density are required.  In this study, we use the

Tsyganenko and Sitnov (2005) magnetic field model and the TWINS exospheric neutral hydrogen density model (Zoennchen, et al, 2015).

We must also deal with the fact that there are two components to the ENA emissions: the energetic ions created in charge exchange interactions with neutral hydrogen in the geocorona, the so-called high altitude emissions (HAE), and those due to charge exchange with neutral oxygen at low altitudes (below ~ 600 km), the so-called low altitude emissions (LAE)  (Roelof,

1997). The former are treated as optically thin emissions, and the latter with a thick target approximation developed by Bazell et al. (2010) and validated by comparisons with DMSP data (Hardy et al., 1984).

A full range of the ion characteristics obtained from the TWINS ENA images have been compared with in-situ measurements.  Measurements of the spatial and temporal variations of the flux in specific energy bands from the Time History of Events and Macroscale Interactions during Substorms (THEMIS) (Angelopoulos, 2008) have been compared with ion flux obtained from the TWINS ENA images (Grimes et al, 2013; Perez et al, 2015).  A similar comparison (Perez et al, 2016) has been made with measurements made on the Van Allen Probes (formerly known as the Radiation Belt Storm Probes (RBSP) A and B) (Mauk et al., 2013; Spence et al.,

2013) by the Radiation Belt Storm Probes Ion Composition Experiment (RBSPICE) (Mitchell et al., 2013) instrument.  Pitch angle distributions and pitch angle anisotropy have been compared with THEMIS observations (Grimes et al, 2013).  Energy spectra have also been compared with

THEMIS measurements (Perez et al, 2012).  Partial pressure and anisotropy from TWINS have been compared with RBSP-SPICE-A (Perez et al, 2016) observations.  While the in-situ measurements show more detailed temporal and spatial features, there is good agreement with the overall trends.  Goldstein et al (2017) compared the TWINS ENA images with in-situ data from THEMIS and the Van Allen probes.  They found evidence for bursty flows and ion structures in the plasma transport during the 2015 St. Patrick's day storm.

**3  The CIMI Model**

The CIMI model is a combination of the Comprehensive Ring Current Model (CRCM) (Fok et al, 2001b) and the Radiation Belt Environment (RBE) model (Fok, et al., 2008).  The CRCM is a combination of the classic Rice Convection Model (RCM) (Harel et al, 1981) and the Fok kinetic model (Fok et al., 1993).

The CRCM simulates the evolution of an inner magnetosphere plasma distribution that conserves the first two adiabatic invariants. The Fok kinetic model solves the bounce-averaged

Boltzmann equation with a specified electric and magnetic field to obtain the plasma distribution.

It is able to include arbitrary pitch angles with a generalized RCM Birkeland current algorithm.

The Fok model advances in time the ring current plasma distribution using either a self- consistent RCM field or the semi-empirical Weimer electric field model.  A specified height- integrated ionospheric conductance is required for the RCM calculation of the electric field.  The

Hardy model (Hardy et al., 1987) provides auroral conductance. Losses along the particle drift paths are a key feature of the CIMI model.  The CIMI pressure distributions utilized in this study cover an energy range from 75 eV to 133 keV.

Simulated results from CIMI or its predecessors have been tested against a variety of measurements from a number of satellite missions.  Some examples are: (1) AMPTE/CCE (Fok et al., 2001b), (2) IMAGE ENA images (Fok et al., 2003), (3) Polar/CEPPAD (Ebihara et al.,

2008), (4) IMAGE/EUV(Buzulukova et al., 2008), (5) TWINS ENA images (Fok, et al., 2010), (6) Radiation belt measurements and Akebono (Glocer, et al..,2011), (7) TWINS plasma sheet boundary conditions (Elfritz, et al., 2014), and (8) TWINS ENA images and Akebono (Fok et al.,

2014).  Using the Dessler-Parker-Schopke relation (Dessler and Parker, 1959; Schokpe, 1966), it has also been shown that the simulated CIMI pressures match well the observed SYM/H. (See

Figure 9, Buzulukova et al., 2010).   In this study, we present the first direct comparison between

CIMI and TWINS ion partial pressure and anisotropy.

Important input to the CIMI simulations are the particles injected into the inner magnetosphere along the outer boundary of the simulation.  In the simulations shown here, it has been assumed that the particles have a Maxwellian distribution with density and temperature determined by a linear relationship to the solar wind density and velocity respectively (Ebihara and Ejiri, 2000; Borovsky et al., 1998). A 2 hour time delay between the arrival of the solar wind parameters at the nose of the magnetopause and its effect on the ions crossing into the inner magnetosphere also has been assumed (Borovsky et al. 1998).  The pitch angle distribution of the incoming ions is taken to be isotropic.

Results from simulations with the CIMI model using two different forms of the electric potential are compared in this investigation.  One is the Weimer 2K empirical model (Weimer,

2001) and the other is a self-consistent electric potential from RCM.

**4 The 7-10 September 2015 Storms**

Figure 1 shows solar wind parameters and geomagnetic activity indices from the OMNI data service for 4 days, i.e., 7-10 September 2015.  During this 4-day period, there were two SYM/H

minima in succession. The first came early on 8 September 2015 after a 1-day long main phase on 7 September 2015.  The minimum SYM/H was approximately -90 nT, so it was a relatively weak storm.  There was a rapid recovery for approximately 3 hours coinciding with a sharp transition of $B_z$ from negative, i.e., -8 or -9 nT, to positive, i.e., +18 or +19 nT along with a sharp transition of $B_y$ from positive, i.e., +5 nT, to negative, i.e., -12 or -13 nT. There was also a sharp spike in the solar wind density at the inception of this first recovery phase. After the recovery was completed, there followed about a 12-hour period of near 0 nT SYM/H. The main phase of the second storm showed a relatively steady decline in SYM/H to a minimum near -110 nT in about 12 hours.  The recovery from this second minimum was slow with a duration of about 1½

days.  The second main phase and minimum corresponded to a slow swing of $B_z$ back to negative and $B_y$ to a slightly negative value.  Also to be noted is the strong AE index, indicative of possible substorm activity during the main phases and early recovery of both minima.  There is also some AE activity near the end of the second storm. During those same periods, the ASY/H

index also had significant values during the main phase and early recovery of both minima. (See

Figure 1.)

**5  Results**

**5.1 Comparison of the Location of the Equatorial Ion Partial Pressure Peaks**

Figure 2 shows the location of the equatorial ion partial pressure peaks as measured from the

TWINS ENA images (green diamonds) and simulated by CIMI with both the Weimer 2K (red lines) and the RCM (orange lines) electric fields.  Figure 2a is the radial location for the four days of the 07-10Sep2015 storms, and Figure 2b is the MLT location.

The radial positions of the partial pressure peaks for the CIMI simulations are similar, i.e., about 4 $R_E$, for both the Weimer 2K and the RCM electric potentials. The RCM results do show more variation.  Many of the radial positions for the TWINS observations are also near 4

$R_E$, but others are at larger values.  The MLT locations of the peaks are generally in the dusk/midnight sector.  This is consistent with statistical analysis of proton fluxes from the database of the magnetospheric plasma analyzer (MPA) instrument aboard Los Alamos satellites at geosynchronous orbit (Korth et al., 1999).  But the CIMI simulations, with both the Weimer

2K and RCM potentials, show a brief time early on 8 September 2015 where some of the peaks are in the midnight/dawn sector.  Given the assumed 2 hour delay in the propagation of the solar wind parameters into the inner magnetosphere, this seems to correlate with a sharp swing in $B_y$

shown in Figure 1.  The TWINS observations show several instances of the partial pressure peaks being near midnight and in the midnight/dawn sector. As described earlier, ion flux peaks in this region have been seen from ENA images for very strong storms (Fok et al, 2003).

**5.2 Comparison of Equatorial Ion Partial Pressure Peaks and Anisotropies at Specific**

**Times**

The following subsections will examine in detail a number of specific times during these two storms in order to address similarities and differences in the simulations with an empirical and a self-consistent electric field model and with observations.  One apparent difference in what follows is the magnitude of the equatorial partial pressure for the three cases.  The maximum on the colorbars for Figures 3-9 were chosen to be different for each time in order to emphasize the spatial dependence of the pressure distribution.  The maxima for the two CIMI simulations are very similar, i.e., the RCM vary from 20-38 nPa and the Weimer 2K from 15-30 nPa.  But the maxima of the TWINS peaks varyfrom 1-4 nPa, which is significantly smaller.

The magnitude of the ion intensities derived from the ENA images has been addressed in several previous comparisons with in-situ measurements.  Vallat et al. (2004) compared Cluster-

CIS (Réme et al., 2001) and IMAGE-HENA observations and found that for relatively strong fluxes, the agreement was excellent for two cases, but for another the ion flux determined from the ENA images was somewhat higher than the in-situ observations and in another it was significantly lower.  Grimes et al. (2013) compared THEMIS (Angleopoulos, 2008) spectral measurements with spectra obtained from TWINS ENA images and found that the in-situ fluxes were a factor of 3 times greater than those obtained from the ENA images.  Perez et al. (2016)

compared 30 keV ion fluxes obtained from TWINS ENA images with in-situ measurements by

RBSPICE-A (Mauk et al., 2013) and found good agreement in both the average time dependent trend and in the magnitude.  The in-situ measurements, of course, showed more structure given their much higher spatial and temporal resolution.  Goldstein et al. (2017) analyzed data from

THEMIS, Van Allen probes, and TWINS for a large storm to find that the ion fluxes obtained from the ENA images were generally lower than those from the in-situ measurements.  They also found significant variations in the in-situ data. So while some part of the difference in the partial pressures obtained from TWINS measurements and CIMI simulations are due to the larger energy range included in the CIMI pressures, it is not the entire explanation. The issue of the absolute magnitude remains an important, unresolved issue, but the fluxes obtained from ENA

images have been shown to reflect the global structure of the trapped ring current particles, and that is the emphasis in this study.

**5.2.1  2200 UT 07 September 2015**

Figure 3 shows the equatorial partial pressure profiles and the pressure anisotropy from the

CIMI/RCM simulation, the TWINS observations, and the CIMI/Weimer 2K simulation at 2200

UT 07 September 2015.  This was late in the main phase of the first storm (See Figure 1.). The radial locations of the peaks differ by less than 1 $R_E$.  The MLT locations of the partial pressure peaks, however, differ by 3 hours in MLT.  While the TWINS peak is near midnight, the CIMI

peaks are well into the dusk/midnight sector with the CIMI/Weimer even closer to dusk. Results for the Weimer96 when compared with the RCM for a very strong storm showed even greater shielding for the RCM when compared to the empirical Weimer model (Fok et al., 2003). Note, however, that for this weaker storm, the MLT spread in the peaks of the partial pressure distributions do overlap.  It is also to be noted that the TWINS results show more radial structure.

The pressure anisotropy shown in Figure 3 is defined as

$$A = \frac{P_\perp - P_\parallel}{P_\perp + P_\parallel}$$

with

$$\begin{Bmatrix} P_\perp \\ P_\parallel \end{Bmatrix} = 2\pi \int_{-1}^{+1} d\cos\alpha \begin{Bmatrix} \sin^2\alpha \\ 2\cos^2\alpha \end{Bmatrix} \
[revised manuscript text omitted]